# SketchThinker-R1: Towards Efficient Sketch-Style Reasoning in Large Multimodal Models

**Ruiyang Zhang**[1,2*]    **Dongzhan Zhou**[2*]    **Zhedong Zheng**[1†]
[1] FST and ICI, University of Macau, China  [2] Shanghai AI Laboratory, China
https://github.com/Ruiyang-061X/SketchThinker-R1

## Abstract

Despite the empirical success of extensive, step-by-step reasoning in large multimodal models, long reasoning processes inevitably incur substantial computational overhead, *i.e.*, in terms of higher token costs and increased response time, which undermines inference efficiency. In contrast, humans often employ sketch-style reasoning: a concise, goal-directed cognitive process that prioritizes salient information and enables efficient problem-solving. Inspired by this cognitive efficiency, we propose *SketchThinker-R1*, which incentivizes sketch-style reasoning ability in large multimodal models. Our method consists of three primary stages. In the *Sketch-Mode Cold Start* stage, we convert standard long reasoning process into sketch-style reasoning and finetune base multimodal model, instilling initial sketch-style reasoning capability. Next, we train *SketchJudge Reward Model*, which explicitly evaluates thinking process of model and assigns higher scores to sketch-style reasoning. Finally, we conduct *Sketch-Thinking Reinforcement Learning* under supervision of SketchJudge to further generalize sketch-style reasoning ability. Experimental evaluation on four benchmarks reveals that our SketchThinker-R1 achieves over 64% reduction in reasoning token cost without compromising final answer accuracy. Qualitative analysis further shows that sketch-style reasoning focuses more on key cues during problem solving.

## 1 Introduction

Since the introduction of Large Reasoning Language Models (LRLMs), such as OpenAI o1 (Jaech et al., 2024) and DeepSeek-R1 (Guo et al., 2025), the deliberate, slow thinking ability has been extensively explored in language models (Zhang et al., 2025; Xu et al., 2025a; Wang et al., 2025d). Inspired by this rapid progress, similar reasoning abilities are being investigated in large multimodal models (Zhou et al., 2025a; Li et al., 2025b; Wang et al., 2025b). These models, through lengthy reasoning procedures, have demonstrated clear improvements across various visual recognition and reasoning tasks (Huang et al., 2025; Yang et al., 2025; Shen et al., 2025a).

However, such extensive reasoning often incurs high token costs (Han et al., 2024) and longer response times (Sui et al., 2025). This inefficiency limits their applicability in real-time scenarios and negatively affects user experience in interactive settings. Moreover, overthinking can harm correctness: redundant steps can introduce misleading information, which could compromise the reasoning efficacy (Chen et al., 2024b) and small errors can accumulate over long chains of reasoning and finally lead to wrong answer (Cuadron et al., 2025). In addition, lengthy reasoning traces are often difficult for humans to interpret, obscuring the core logic behind predictions.

In contrast, sketching, a natural human behavior, offers both efficiency and effectiveness in problem-solving (Cross, 2006). By quickly writing down the essential steps and logic, humans reach accurate solutions. Notably, sketch-style reasoning focuses only on the critical steps, keeping the process concise while still leading to correct answers. Inspired by this cognitive behavior, we investigate whether similar reasoning pattern can be developed in large multimodal model to improve its reasoning efficiency without sacrificing accuracy.

In this paper, we propose *SketchThinker-R1*, a reinforcement learning framework which incentivizes sketch-style reasoning in large multimodal models to facilitate reasoning efficiency. Our approach

---

*Equal contribution. †Correspondence to zhedongzheng@um.edu.mo.

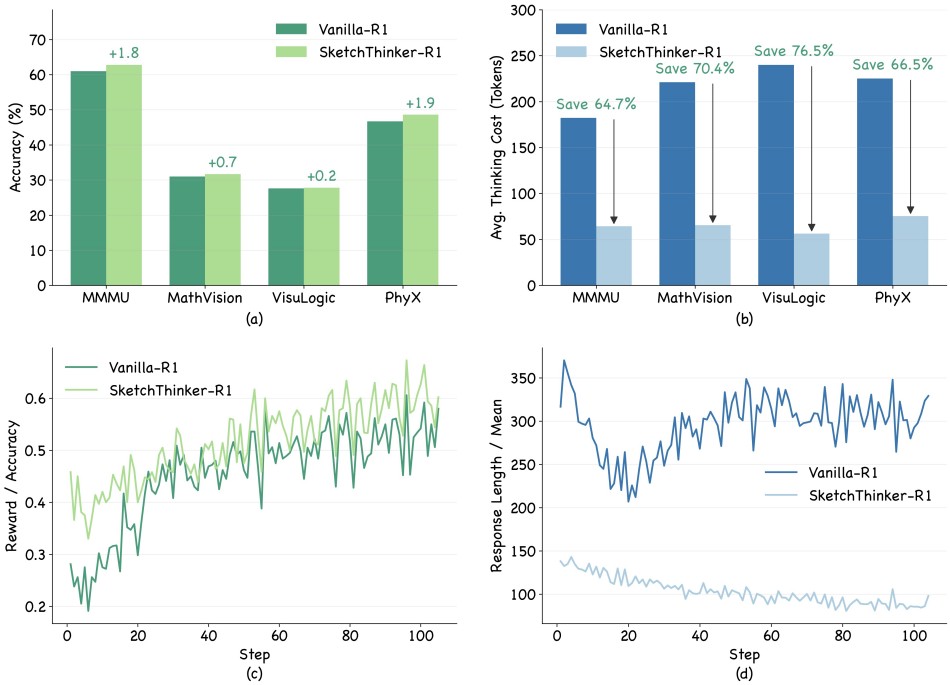

Figure 1: Our SketchThinker-R1 significantly reduces the thinking cost without compromising final answer accuracy. Vanilla-R1 serves as the baseline, representing the standard R1-style trained model. Evaluation across four benchmarks from diverse domains shows that our model achieves comparable or even superior performance (see (a)). At the same time, it reduces thinking token cost by more than 64% (see (b)). During RL training, sketch-style reasoning consistently yields higher accuracy rewards (see (c)) while maintaining a much shorter reponse length (see (d)).

encourages models to generate reasoning traces that preserve only the key logical flow required to solve a problem, thereby reducing thinking cost while maintaining accuracy (see Fig. 1). The framework consists of three stages: (1) **Sketch-Mode Cold Start.** We construct sketch-style reasoning data and perform supervised fine-tuning on base multimodal models to instill initial sketch reasoning ability. Specifically, we leverage existing multimodal reasoning datasets that contain detailed, long-form reasoning. Utilizing strong proprietary LLM, we convert thinking style of these longform reasoning into concise sketch-style, preserving essential logic and removing redundant details. (2) **SketchJudge Reward Model.** In this stage, we utilize the two-mode thinking data (sketch-style and normal reasoning) from cold start stage to further train an LLM to become SketchJudge reward model. This SketchJudge reward model explicitly evaluates reasoning traces, assigning higher scores to concise, sketch-style reasoning while penalizing unnecessarily verbose explanations. It provides a reliable supervisory signal for the following reinforcement learning process. (3) **Sketch-Thinking Reinforcement Learning.** Finally, we perform reinforcement learning on the cold-started model under the guidance of SketchJudge to further generalize the sketch-thinking ability. Our reward design explicitly incorporates evaluation from SketchJudge, encouraging models to produce sketch-style, concise reasoning traces. The reinforcement learning is conducted on datasets across diverse domains to ensure the generality and robustness of the learned sketch-thinking capability. We evaluate *SketchThinker-R1* across four benchmarks spanning different domains. Our model achieves over 64% reduction in thinking token cost, while maintaining and even improving final answer accuracy. Qualitative analysis further reveals that sketch-style reasoning focuses on the key logical steps necessary for problem-solving, offering both efficiency and interpretability. Our main contributions are summarized as follows:

- **Sketch-based Reasoning Framework.** We introduce *SketchThinker-R1*, a novel reinforcement learning framework that fine-tunes Large Multimodal Models (LMMs) to produce concise, sketch-like reasoning chains. By directly rewarding brevity and correctness, our method guides the model to distill complex reasoning into its most essential steps.

- **State-of-the-Art Reasoning Efficiency.** Extensive experiments on four multimodal reasoning benchmarks validate that *SketchThinker-R1* reduces the token cost of the reasoning process by over 64% with no degradation in final answer accuracy, setting a new benchmark for efficient multimodal reasoning.

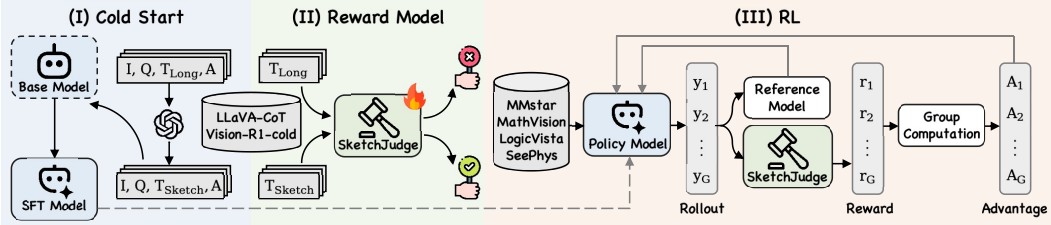

Figure 2: **Overview of our SketchThinker-R1 pipeline.** (1) In the Sketch-Mode Cold Start stage, we convert long reasoning processes from existing multimodal reasoning datasets into sketch-style, and fine-tune the base multimodal model to instill initial sketch-style reasoning ability. (2) Next, we train a SketchJudge Reward Model, which favors sketch-style reasoning and penalizes overly verbose reasoning. (3) Finally, we perform Sketch-Thinking Reinforcement Learning on the cold-started multimodal model under the supervision of the trained SketchJudge reward model, further enhancing the sketch-thinking ability.

## 2 METHOD

### 2.1 SKETCH-MODE COLD START

The first stage of our SketchThinker-R1 framework is the sketch-mode cold start. This stage aims to instill initial sketch-style thinking ability into base multimodal model, laying a solid foundation for the subsequent reinforcement learning generalization process. Specifically, our data source is based on two multimodal reasoning datasets, *e.g.*, LLaVA-CoT-100K (Xu et al., 2024) and Vision-R1-cold (Huang et al., 2025). These datasets already contain images $I$, questions $Q$, long chain-of-thought reasoning $T_{Long}$, and answers $A$. Based on such multimodal reasoning data, we convert the long, detailed reasoning process $T_{Long}$ into a sketch-style reasoning process $T_{Sketch}$. We leverage strong LLM to conduct this transformation. (1) The first aspect is minimizing the content while keeping the key logical flow. Irrelevant details and verbose explanations are removed, along with specific examples. (2) The second aspect is converting the reasoning into a numbered list format, with key steps separated and clearly listed. The full prompt for the sketch-style reasoning data generation is provided in Appendix A. The supervised fine-tuning process minimizes the following objective across all training samples:

$$\mathcal{L}_{\text{SFT}} = -\frac{1}{N} \sum_{i=1}^{N} \sum_{t=1}^{T_i} \log \pi_\theta \big( o_{i,t} \mid o_{i,<t}, q_i \big). \tag{1}$$

where $N$ is the number of training samples, $o_{i,t}$ denotes the $t$-th token of the output sequence for the $i$-th sample, $T_i$ is the total length of the output sequence of the $i$-th training sample, $o_{i,<t}$ represents the tokens preceding the $t$-th token of the $i$-th training sample, $q_i$ is the input query for the $i$-th training sample, and $\pi_\theta$ is the model policy parameterized by $\theta$.

**Discussions. Why not directly apply reinforcement learning to incentivize sketch-style thinking?** We find that letting base multimodal model directly explore sketch-style reasoning leads to quite slow learning process. As a result, the eventual reduction in reasoning cost is only marginal. In contrast, performing sketch-mode cold start on the base model explicitly instills initial sketch-style reasoning ability, enabling more effective learning and exploration during the reinforcement learning process. The sketch-mode cold start stage substantially contributes to improvement of reasoning efficiency. **What are the characteristics of sketch-style reasoning?** Sketch-style reasoning is substantially shorter than long-form reasoning. It preserves only the essential logical flow in an abstract form. Consequently, cold-start training with sketch-style reasoning data can foster highly efficient reasoning ability. At the same time, sketch-style reasoning retains all the key cues required to solve the problem, thereby ensuring strong problem-solving capability.

## 2.2 SKETCHJUDGE REWARD MODEL

The second stage of our framework involves building the SketchJudge reward model, which explicitly evaluates reasoning style of the model. It favors sketch-style reasoning and penalizes long, detailed reasoning. This reward model is then used to supervise the subsequent reinforcement training process, guiding the model to generalize sketch-style thinking. In practice, we fine-tune an open-source LLM for this purpose. Specifically, we leverage both normal reasoning data $T_{Long}$ and sketch-style reasoning data $T_{Sketch}$ from the cold-start stage to construct the fine-tuning dataset. Normal reasoning is assigned a score of 0, while sketch-style reasoning is assigned a score of 1. The template for building the SketchJudge fine-tuning data is as follows: "Give a score of 1 for sketch-style thinking and a score of 0 for normal thinking. Normal thinking contains detailed analysis. Sketch-style thinking contains only the key logic flow. Only output the final score. Now, score this thinking process: [THINKING]". We utilize the same prompt for SketchJudge to evaluate the thinking process during the subsequent reinforcement learning stage.

**Discussions. Why conduct supervised fine-tuning on the LLM to build SketchJudge?** Performing supervised fine-tuning on the base LLM for reasoning-style evaluation can improve evaluation accuracy. Precise thinking-style reward signals make reinforcement learning more effective, leading to both reduced reasoning length and improved final answer accuracy. In contrast, directly prompting an off-the-shelf LLM to evaluate reasoning style yields less reliable reward signals. Such noisy rewards can ultimately compromise the effectiveness of the reinforcement learning process. **Why explicitly supervise the style of the reasoning?** Our SketchJudge reward model only poses supervision on the style of the reasoning process, with no strict restriction on the thinking length. This design can lead to more adaptive thinking behavior. Our SketchThinker-R1 can still generate relatively long and extensive reasoning for challenging questions. Meanwhile, because the thinking style is strictly controlled, the average thinking cost is still significantly reduced.

## 2.3 SKETCH-THINKING REINFORCEMENT LEARNING

The final stage of our SketchThinker-R1 framework is sketch-thinking reinforcement learning. This process is guided by the trained SketchJudge reward model and aims to generalize the sketch-style thinking ability established during the sketch-mode cold start stage. In practice, we adopt the off-the-shelf Group Reward Proximal Optimization (GRPO) algorithm (Shao et al., 2024). Specifically, GRPO performs multiple rollout samplings and optimizes the policy to favor responses with higher assigned rewards, the training objective of GRPO is as follows:

$$J_{\text{GRPO}}(\theta) = \mathbb{E}_{q \sim P(Q), \{o_i\}_{i=1}^G \sim \pi_{\theta_{\text{old}}}(O|q)}$$
$$\left[ \frac{1}{G} \sum_{i=1}^G \min\left( \frac{\pi_\theta(o_i \mid q)}{\pi_{\theta_{\text{old}}}(o_i \mid q)} A_i, \ \text{clip}\left( \frac{\pi_\theta(o_i \mid q)}{\pi_{\theta_{\text{old}}}(o_i \mid q)}, \ 1 - \epsilon, \ 1 + \epsilon \right) A_i \right) - \beta \, D_{KL}(\pi_\theta \,\|\, \pi_{\text{ref}}) \right], \tag{2}$$

$$\mathbb{D}_{KL}(\pi_\theta \,\|\, \pi_{\text{ref}}) = \frac{\pi_{\text{ref}}(o_i \mid q)}{\pi_\theta(o_i \mid q)} - \log \frac{\pi_{\text{ref}}(o_i \mid q)}{\pi_\theta(o_i \mid q)} - 1, \tag{3}$$

where $\pi_\theta$ is the current model policy, $\pi_{\theta_{\text{old}}}$ is the old policy, $G$ is the rollout group size, $q$ is the query, $o_i$ is the $i$-th sampled reponse, $\pi_{\text{ref}}$ is the reference model, $\epsilon$ is the clipping hyper-parameter controlling updating degree, and $\beta$ is the coefficient of Kullback–Leibler (KL) penalty.

$$A_i = \frac{r_i - \text{mean}(\{r_1, r_2, \cdots, r_G\})}{\text{std}(\{r_1, r_2, \cdots, r_G\})}, \tag{4}$$

$A_i$ is the normalized advantages computed based on rewards $\{r_1, r_2, \cdots, r_G\}$.

Our thinking style reward shaping is as follows:

$$R_i = 0.5 \times R_{\text{accuracy}}(o_i) + 0.4 \times R_{\text{format}}(o_i) + 0.1 \times R_{\text{thinking-style}}(o_i), \tag{5}$$

where $R_{\text{accuracy}}(o_i)$ evaluate answer accuracy, $R_{\text{format}}(o_i)$ check whether response match the required response format, $R_{\text{thinking-style}}(o_i)$ is the thinking-style reward score assigned by our SketchJudge.

The reasoning process is parsed from the final response of the model and fed into the SketchJudge to obtain this thinking-style reward:

$$R_{\text{thinking-style}}(o_i) = \begin{cases} 1 & \text{if SketchJudge checks the thinking process as sketch-style,} \\ 0 & \text{if SketchJudge checks the thinking process as normal-style.} \end{cases} \tag{6}$$

## 3 EXPERIMENT

**Dataset.** **SketchColdStart-20K.** We construct our cold start dataset based on LLaVA-CoT-100K (Xu et al., 2024) and Vision-R1-cold (Huang et al., 2025), both of which are multimodal reasoning datasets that already contain long reasoning processes. We randomly sample 10K questions from LLaVA-CoT-100K and another 10K questions from Vision-R1-cold, resulting in a total of 20K questions. The long reasoning processes are extracted from the full model responses and then converted into sketch-style reasoning utilizing GPT-5. The detailed conversion prompt is provided in the Appendix A. **SketchRL-1K.** To construct the RL training set, we draw from four data sources across different domains to enhance the generalization of sketch-style reasoning ability. MMStar (Chen et al., 2024a) contains 1,500 multiple-choice questions for general visual understanding tasks. MathVista (Lu et al., 2023) focuses on mathematical visual reasoning, including both multiple-choice and free-form questions, with 5,140 samples in total. LogicVista (Xiao et al., 2024) includes 448 samples targeting various visual logic skills. SeePhys (Xiang et al., 2025) contains 2,000 free-form samples of visual physics questions. From each source, we randomly sample 250 questions, forming a final RL training set of 1,000 samples.

**Implementation Details.** For RL training, we utilize Easy-R1 (Zheng et al., 2025). We employ GRPO (Shao et al., 2024) for training. The maximum prompt length is set to 2048 and maximum response length is 2048. The KL coefficient is set to 0.01. We utilize AdamW as optimizer, with learning rate of 1e-6 and weight decay of 1e-2. For rollout, we set sampling time to 5 and sampling temperature to 1.0. Rollout batch size is set to 128. We train for 15 epochs, resulting in 105 training steps. For supervised fine-tuning during Sketch-Mode Cold Start and SketchJudge reward model training, we leverage LLaMA-Factory (Zheng et al., 2024). We use 8 H200 for all our experiments.

**Evaluation.** We conduct evaluations on four multimodal benchmarks from various domains to ensure a comprehensive assessment of our SketchThinker-R1. MMMU (Yue et al., 2024) is a comprehensive benchmark designed to evaluate the general reasoning ability of large multimodal models, covering questions from a wide range of topics and includes 100, 900, and 10,500 questions in the dev, validation, and test sets. MathVision (Wang et al., 2024) specifically benchmarks mathematical visual question solving ability, containing both free-form and multiple-choice questions, with 3,040 questions in test set and testmini set of 304 questions. VisuLogic (Xu et al., 2025c) primarily focuses on benchmarking logical reasoning ability, with particular emphasis on understanding visual information, which contains 1,000 samples. PhyX (Shen et al., 2025b) is a recently proposed benchmark aimed at evaluating capability of model in visual physics question understanding, which consists of 3,000 visually grounded physics questions.

**Baselines.** Vanilla-R1: We perform vanilla R1-style training (Guo et al., 2025) on the base multimodal model with the same training data as SketchThinker-R1 to establish this baseline. Constrained CoT (Nayab et al., 2024): Directly prompt Vanilla-R1 to restrict the reasoning process within a specified word count. Chain-of-Draft (Xu et al., 2025b): Prompt Vanilla-R1 to constrain each reasoning step to a certain word count. C3oT (Kang et al., 2025): Mix both short CoT and long CoT data to finetune the base multimodal model. VeriThinker (Chen et al., 2025): Construct short-CoT data using a small non-reasoning model, then finetune the base multimodal model on this data for a verification task to build efficient reasoning ability. L1 (Aggarwal & Welleck, 2025): Conduct reinforcement learning with a length-based reward to control the reasoning length, where the response length is compared with a fixed golden length, and the L1 difference is used as the length reward. ThinkPrune (Hou et al., 2025): First truncate the model response to a fixed target length, then calculate the accuracy reward based only on the truncated reponse.

**Metrics.** We adopt three metrics for evaluation. Acc. denotes answer accuracy. #Token represents the average token count of model reasoning process. In addition, we define a new metric, Efficiency of Thinking (EoT), to explicitly measure the efficiency of reasoning. EoT is calculated as $\frac{Acc}{N_{token}}$.

| Method | MMMU | | | MathVision | | | VisuLogic | | | PhyX | | |
|---|---|---|---|---|---|---|---|---|---|---|---|---|
| | Acc.↑ | #Token↓ | EoT↑ | Acc.↑ | #Token↓ | EoT↑ | Acc.↑ | #Token↓ | EoT↑ | Acc.↑ | #Token↓ | EoT↑ |
| *Direct Inference* | | | | | | | | | | | | |
| Vanilla-R1 | 61.0 | 182.2 | 0.335 | 31.0 | 221.1 | 0.140 | 27.6 | 240.0 | 0.115 | 46.7 | 225.1 | 0.207 |
| *Prompt-based* | | | | | | | | | | | | |
| Constrained CoT | 58.6 | 78.2 | 0.749 | 26.2 | 79.2 | 0.331 | 26.4 | 71.5 | 0.369 | 42.4 | 73.4 | 0.578 |
| Chain-of-Draft | 58.9 | 86.3 | 0.683 | 27.4 | 85.4 | 0.321 | 26.5 | 94.6 | 0.280 | 42.2 | 85.2 | 0.495 |
| *Supervised-fine-tuning-based* | | | | | | | | | | | | |
| C3oT | 59.3 | 127.1 | 0.467 | 28.8 | 125.5 | 0.229 | 27.1 | 134.6 | 0.201 | 43.8 | 117.5 | 0.373 |
| VeriThinker | 60.1 | 105.8 | 0.568 | 29.1 | 152.5 | 0.191 | 27.5 | 107.4 | 0.256 | 45.5 | 132.7 | 0.343 |
| *Reinforcement-Learning-based* | | | | | | | | | | | | |
| L1 | 59.5 | 136.8 | 0.435 | 29.5 | 146.7 | 0.201 | 27.2 | 167.4 | 0.162 | 45.1 | 153.4 | 0.294 |
| ThinkPrune | 59.2 | 104.9 | 0.564 | 29.6 | 136.3 | 0.217 | 26.9 | 183.2 | 0.147 | 46.3 | 163.7 | 0.283 |
| **SketchThinker-R1-7B** | **62.8** | **64.3** | **0.977** | **31.7** | **65.5** | **0.484** | **27.8** | **56.3** | **0.494** | **48.6** | **75.3** | **0.645** |

Table 1: **Comparison with state-of-the-art efficient thinking methods.** Quantitative results across four benchmarks from different domains validate the thinking efficiency of our SketchThinker-R1-7B. Compared with various baselines, including prompt-based, supervised fine-tuning, and reinforcement learning methods, our model achieves higher accuracy with fewer reasoning tokens. We utilize Qwen2.5-VL-7B-Instruct as backbone for both our method and all baselines for fair comparison. Vanilla-R1 refers to results of standard R1-style trained model.

| Method | MMMU | | | MathVision | | | VisuLogic | | | PhyX | | |
|---|---|---|---|---|---|---|---|---|---|---|---|---|
| | Acc.↑ | #Token↓ | EoT↑ | Acc.↑ | #Token↓ | EoT↑ | Acc.↑ | #Token↓ | EoT↑ | Acc.↑ | #Token↓ | EoT↑ |
| *Direct Inference* | | | | | | | | | | | | |
| Vanilla-R1 | 54.8 | 128.3 | 0.427 | **26.9** | 151.2 | 0.178 | 25.6 | 139.5 | 0.184 | 34.8 | 173.2 | 0.201 |
| *Prompt-based* | | | | | | | | | | | | |
| Constrained CoT | 52.7 | 76.2 | 0.692 | 22.1 | 63.4 | 0.349 | 25.2 | 69.1 | 0.365 | 31.5 | 63.6 | 0.495 |
| Chain-of-Draft | 53.8 | 72.1 | 0.746 | 22.7 | 71.5 | 0.317 | 25.1 | 74.2 | 0.338 | 32.5 | 71.2 | 0.456 |
| *Supervised-fine-tuning-based* | | | | | | | | | | | | |
| C3oT | 54.1 | 107.5 | 0.503 | 24.1 | 105.1 | 0.229 | 25.1 | 104.3 | 0.241 | 33.1 | 93.2 | 0.355 |
| VeriThinker | 52.2 | 95.8 | 0.545 | 23.1 | 127.6 | 0.181 | 25.2 | 93.7 | 0.269 | 32.6 | 92.1 | 0.354 |
| *Reinforcement-Learning-based* | | | | | | | | | | | | |
| L1 | 53.7 | 102.6 | 0.523 | 24.7 | 107.0 | 0.231 | 25.2 | 97.6 | 0.258 | 32.2 | 83.4 | 0.386 |
| ThinkPrune | 53.2 | 95.2 | 0.559 | 23.8 | 92.3 | 0.258 | 25.4 | 73.3 | 0.347 | 33.2 | 82.2 | 0.404 |
| **SketchThinker-R1-3B** | **55.9** | **54.5** | **1.026** | 25.3 | **72.7** | **0.348** | **25.8** | **36.9** | **0.699** | **35.1** | **67.3** | **0.522** |

Table 2: **Quantitative comparison between SketchThinker-R1-3B and other baselines.** Our 3B model consistently outperforms various baselines in both answer accuracy and reasoning token cost. This validates the robustness of our proposed framework in eliciting efficient reasoning across models of different scales. All methods utilize Qwen2.5-VL-3B-Instruct as backbone.

## 3.1 MAIN RESULTS

We present quantitative comparison between SketchThinker-R1-7B and various efficient thinking baselines (see Tab. 1). We observe that our SketchThinker-R1-7B surpasses various methods by clear margin in both final answer accuracy and thinking token cost. This validates the effectiveness of our proposed sketch-style thinking training pipeline, including sketch-mode cold start and sketch-thinking reinforcement learning. For prompt-based efficient thinking methods, the thinking process is forcibly constrained, leading to insufficient reasoning and significantly compromising final answer accuracy. For SFT-based efficient thinking methods, the ability is primarily fit to the training dataset, resulting in sub-optimal performance when generalizing to out-of-domain benchmarks. In contrast, our SketchThinker-R1 training framework builds inherent sketch-thinking ability into model, enabling it to solve questions with concise and effective reasoning. Our model achieves

(a)

| Sketch-Mode Cold Start | Sketch-Thinking RL | Acc.↑ | #Token↓ | EoT↑ |
|---|---|---|---|---|
| ✓ | | 61.4 | 114.5 | 0.536 |
| | ✓ | 62.1 | 152.2 | 0.408 |
| ✓ | ✓ | **62.8** | **64.3** | **0.977** |

(c)

| Cold Start Data Source | Acc.↑ | #Token↓ | EoT↑ |
|---|---|---|---|
| LLaVA-CoT-100K | 61.9 | 68.5 | 0.904 |
| Vision-R1-cold | 61.2 | 71.2 | 0.860 |
| LLaVA-CoT-100K & Vision-R1-cold | **62.8** | **64.3** | **0.977** |

(b)

| Model | Closed-Source | Acc.↑ | #Token↓ | EoT↑ |
|---|---|---|---|---|
| GPT-5 | ✓ | **62.8** | 64.3 | **0.977** |
| o4-mini | ✓ | 62.2 | 65.3 | 0.953 |
| GPT-OSS-20B | | 60.5 | 63.2 | 0.957 |
| Qwen2.5-72B-Instruct | | 59.1 | **62.3** | 0.949 |

(d)

| SketchJudge Reward Model | SFT | Acc.↑ | #Token↓ | EoT↑ |
|---|---|---|---|---|
| Qwen2.5-7B-Instruct | ✓ | **62.8** | **64.3** | **0.977** |
| Qwen2.5-7B-Instruct | | 61.0 | 72.1 | 0.846 |
| Qwen2.5-3B-Instruct | ✓ | 59.8 | 74.8 | 0.799 |
| Qwen2.5-3B-Instruct | | 59.4 | 78.1 | 0.761 |

Table 3: (a) Ablation study of our primary stages. We observe that combining both sketch-mode cold start and sketch-thinking RL achieves the highest thinking efficiency. Applying only sketch-mode cold start mainly instills sketch-style reasoning ability fitted to the training data, resulting in limited performance on test benchmarks. In contrast, applying sketch-thinking RL without cold start leads to ineffective exploration and learning, yielding only marginal reductions in thinking cost. (b) Ablation study of the LLM used for converting long reasoning into sketch-style reasoning. We observe that cold-start data generated by GPT-5 yields optimal thinking efficiency. Therefore, we adopt GPT-5–generated data to build SketchColdStart-20K. Data from open-source LLMs reduce token cost more aggressively, but at the expense of lower answer accuracy. (c) Ablation study of data sources for constructing cold-start data. Combining diverse data sources simultaneously improves answer accuracy and reduces reasoning cost. Leveraging multiple sources enhances robustness and fosters more general sketch-style reasoning ability during the cold-start stage. (d) Ablation study of the SketchJudge reward model. Qwen2.5-7B-Instruct, after fine-tuning for scoring ability, achieves the best thinking efficiency. Providing more accurate supervision signals during reinforcement learning leads to a more stable and effective training process, thereby enhancing the final sketch-style reasoning ability. Best results are **bolded**. Gray line is our default setting. All ablation results are from MMMU. The utilized backbone is Qwen2.5-VL-7B-Instruct.

optimal thinking efficiency across various benchamrks. Compared with normally trained R1 model, our model attains comparable answer accuracy with over 64% savings in thinking cost.

We also present the quantitative comparison results of our 3B model (see Tab. 2). For the 3B scale, our framework also effectively elicits highly efficient reasoning ability. Compared with various baselines, including prompt-based, SFT-based, and reinforcement learning–based methods, our approach achieves higher answer accuracy while requiring fewer reasoning tokens. This demonstrates the general effectiveness of our method in improving reasoning efficiency across different model scales. Notably, compared with Vanilla-R1, our method reduces thinking cost by more than 50% while maintaining comparable answer accuracy. Compared with the 7B model, the reduction rate of thinking cost is slightly lower. This is because the 3B model generally produces shorter reasoning chains with less redundancy, leaving lesser room for efficiency improvement.

## 3.2 Ablation Study and Analysis

**Ablation of primary stages.** We present an ablation study of our primary stages, including sketch-mode cold start and sketch-thinking reinforcement learning (see Tab. 3a). We observe that combining sketch-mode cold start with sketch-thinking reinforcement learning achieves both the highest reasoning efficiency and the best answer accuracy. Relying solely on sketch-mode cold start build sketch-thinking ability that is largely restricted to the training set, resulting in suboptimal performance on test benchmarks. Conversely, applying sketch-thinking reinforcement learning without initializing sketch-style ability from cold start leads to ineffective exploration and yields only marginal reductions in thinking cost.

**Ablation of the LLM used for SketchColdStart-20K construction.** We report ablation results of different LLMs utilized to extract sketch-style reasoning from long reasoning traces during construction of SketchColdStart-20K (see Tab. 3b). We find that cold-start data generated by GPT-5 leads

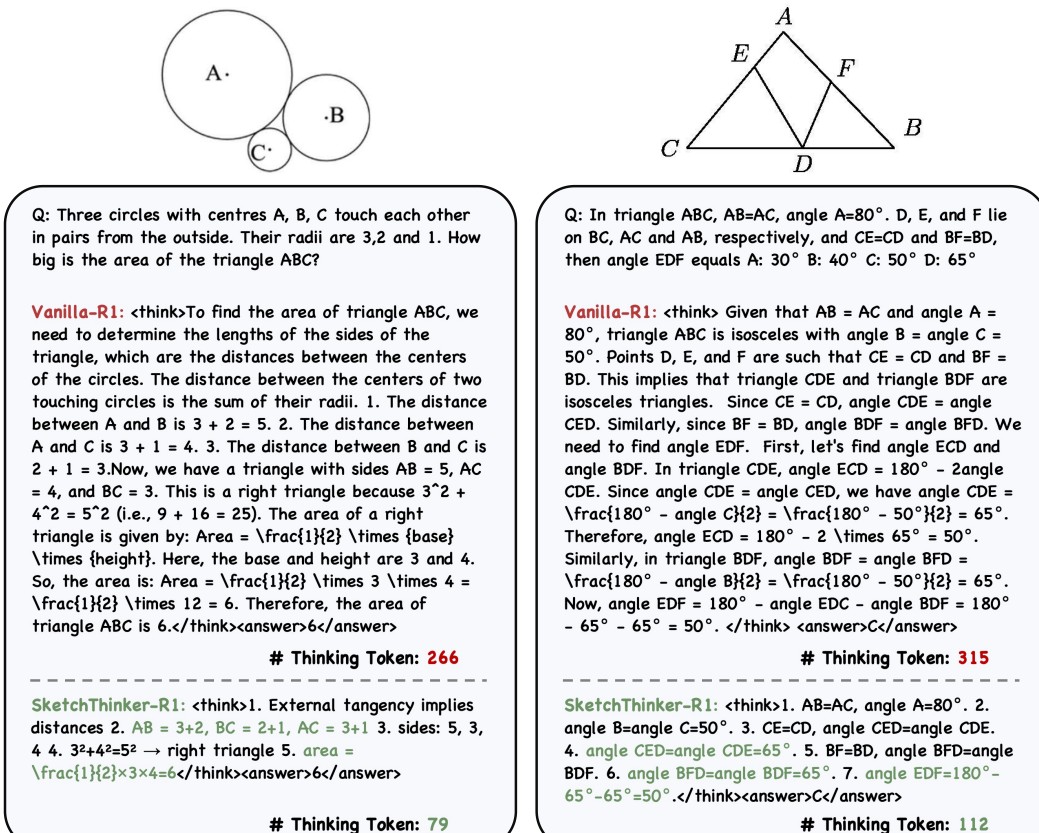

Figure 3: **Qualitative analysis of our SketchThinker-R1.** SketchThinker-R1 conducts a highly efficient yet effective sketch-style reasoning process. By focusing on key cues in problem-solving, our model arrives at the correct answer. The samples are from MathVision (Wang et al., 2024).

to the most effective development of sketch-style thinking ability. Models cold-started with data from closed-source LLMs achieve relatively higher final answer accuracy, because closed-source models tend to produce more reliable sketch-style reasoning processes, which in turn improves accuracy after reinforcement training. In contrast, cold-start data generated by open-source LLMs results in more efficient reasoning ability, as these models tend to produce more concise sketch-style reasoning. Overall, distribution of cold-start data significantly influences final learned ability after reinforcement learning, shaping whether the trained model emphasizes accuracy or efficiency.

**Ablation of the cold start data source.** We present the influence of data source used to construct SketchColdStart-20K (see Tab. 3c). We observe that combining multiple data sources simultaneously improves both final answer accuracy and reasoning efficiency. A more diverse set of cold-start data enables the model to acquire a broader and more general sketch-thinking ability without overfitting to a single distribution. During the subsequent reinforcement learning stage, this broader capability can be more effectively generalized, resulting in further gains in reasoning efficiency.

**Ablation of SketchJudge reward model.** We present results of different SketchJudge reward models (see Tab. 3d). We observe that Qwen2.5-7B-Instruct, after scoring ability fine-tuning, enables SketchThinker-R1 to achieve optimal thinking efficiency. Providing a more reliable reward signal during reinforcement learning enables the model to more effectively learn sketch-style reasoning ability. Supervised fine-tuning enhances the scoring accuracy of the SketchJudge reward model, thereby improving reward assignment and guiding the learning of more effective sketch-style reasoning. In addition, employing a larger backbone improves the evaluation of the model reasoning process during reinforcement learning, leading to stronger overall sketch-thinking ability.

**Qualitative cases.** We also present qualitative examples of our SketchThinker-R1 (see Fig. 3). We observe that SketchThinker-R1 generates reasoning processes that concentrate on the key cues

required to solve the problem, leading to a highly efficient thinking process. Compared with Vanilla-R1, thinking cost of SketchThinker-R1 is significantly lower, while still producing correct answers.

## 4 RELATED WORK

**Large Multi-modal Reasoning Model.** Driven by the success of RL in eliciting reasoning from LLMs (Guo et al., 2025; Jaech et al., 2024), researchers have attempted to replicate this success in the multimodal domain (Li et al., 2025b; Wang et al., 2025b). VisualThinker-R1-Zero (Zhou et al., 2025b) and MM-Eureka (Meng et al., 2025) pioneer the exploration of spatial reasoning and mathematical VQA tasks, successfully reproducing long CoT characteristics in large multimodal models. Taking one step further, OThink-MR1 (Liu et al., 2025) propose dynamic weighting of the KL divergence term to enhance GRPO (Shao et al., 2024), balancing exploration in early RL stages with exploitation in later stages. ThinkLite-VL (Wang et al., 2025a) introduce difficulty quantification for training samples, measuring the iterations required to solve each problem and selecting samples of appropriate difficulty for RL training. Building on these early efforts, Vision-R1 (Huang et al., 2025) leverage VLMs and DeepSeek-R1 (Guo et al., 2025)to construct reasoning data, initialize reasoning ability through cold-start training, and propose Pprogressive Thinking Suppression Training (PTST) to incrementally increase reasoning length. LMM-R1 (Peng et al., 2025) adopt a two-stage pipeline, first leveraging large-scale text-only data to strengthen reasoning before transitioning to multimodal reasoning. Beyond text-image reasoning, some works extend to other modalities (Xie et al., 2025; Zhang et al., 2024a;b). R1-Omni (Zhao et al., 2025) conduct preliminary attempts on video and audio tasks, while Spatial-R1 (Ouyang, 2025) extend R1-style training to video spatial reasoning. However, these models generally exhibit long reasoning processes, resulting in low efficiency. In contrast, *SketchThinker-R1* explicitly optimizes the reasoning process while preserving answer accuracy by building sketch-style reasoning ability in large multimodal models.

**Efficient Reasoning.** Research on efficient reasoning has mainly followed three trajectories (Feng et al., 2025; Sui et al., 2025). The first is *prompt-based efficient reasoning*, which controls reasoning length through instructions. For example, TALE-EP (Han et al., 2024) prompts models to estimate the required token budget for a given question, then constrains reasoning within that budget. Chain-of-Draft (Xu et al., 2025b) further controls the length of each reasoning step through prompt design. The second trajectory is *training-based efficient reasoning*, which directly trains models to produce concise reasoning. These methods typically involve collecting short reasoning data and conducting SFT. For instance, Self-Training (Munkhbat et al., 2025) samples multiple outputs for the same question, selecting the shortest reasoning to construct a training set. TokenSkip (Xia et al., 2025) estimates the semantic importance of each reasoning segment and prunes unnecessary tokens. The third trajectory is *output-based efficient reasoning*, which reduces reasoning length dynamically at inference. Several works replace explicit CoT with latent representations. Coconut (Hao et al., 2024) treats the final hidden states of an LLM as continuous thoughts, bypassing tokenized reasoning. CODI (Shen et al., 2025c) extends this by training models to learn continuous latent CoT via self-distillation. Other works explore sampling-based methods to improve efficiency without retraining (Li et al., 2025a; Wang et al., 2025c). Different from existing works, *SketchThinker-R1* explicitly supervises the thinking style of model during reinforcement learning through the SketchJudge reward model, thereby enhancing reasoning efficiency by fostering sketch-style thinking ability.

## 5 CONCLUSION

In this paper, we propose SketchThinker-R1, a reinforcement learning framework designed to incentivize efficient sketch-style reasoning in large multimodal models. Our framework consists of three main stages: (1) Sketch-Mode Cold Start. We construct SketchColdStart-20K, a sketch-style reasoning dataset created by converting long reasoning processes from diverse data sources to sketch-style. By fine-tuning on this dataset, base multimodal models are endowed with initial sketch-style reasoning ability. (2) SketchJudge Reward Model. We develop a reward model that explicitly evaluates the reasoning process and assigns higher scores to concise sketch-style thinking. (3) Sketch-Thinking Reinforcement Learning. We apply reinforcement learning under the supervision of SketchJudge to further generalize and strengthen the sketch-style reasoning ability. Extensive evaluations on four benchmarks across different domains illustrate the thinking efficiency of SketchThinker-R1, which achieves over 64% reduction in reasoning length while maintaining comparable answer accuracy.

## ACKNOWLEDGMENT

We acknowledge supports from Guangdong Basic and Applied Basic Research Foundation 2025A1515012281, Nanjing Municipal Science and Technology Bureau 202401035 and University of Macau MYRG-GRG2024-00077-FST-UMDF.

## ETHICS STATEMENT

In this work, we investigate incentivizing sketch-style reasoning ability in large multimodal models to improve reasoning efficiency. The training data used in both the cold-start and reinforcement learning stages are drawn entirely from publicly available sources and contain no harmful or sensitive content. Our research focuses exclusively on improving the reasoning efficiency of large multimodal models, without involving human subjects, personal data, or safety-critical applications.

## REPRODUCIBILITY STATEMENT

We provide a detailed presentation of our implementation process in Sec. 3 and Appendix D. The construction procedures of both SketchColdStart-20K and SketchRL-1K are described in detail. All key hyperparameters used in supervised fine-tuning and reinforcement training are reported. In addition, we include the full prompts employed for sketch-thinking data construction. All code, models, and datasets will be released to ensure reproducibility.

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

APPENDIX

## A   PROMPT DESIGN

We present the prompt used for sketch-style reasoning generation (see Tab. 16). The LLM is instructed to retain the key logic for problem solving while removing unnecessary details. In addition, it is prompted to structure the reasoning process into a numbered list, making the reasoning clearer and easier to follow.

## B   COLD START DATA ILLUSTRATION

We provide several examples from our cold-start dataset SketchColdStart-20K (see Fig. 4). The generated sketch-style reasoning process is highly concise. Fine-tuning the base multimodal model on SketchColdStart-20K establishes a solid foundation for developing efficient reasoning ability.

## C   QUALITATIVE CASES

We present qualitative cases of SketchThinker-R1 (see Fig. 5, Fig. 6, Fig. 7, Fig. 8). SketchThinker-R1 generates substantially shorter reasoning processes while still arriving at correct answers. Its sketch-style reasoning emphasizes key cues for solving the given questions, resulting in a much more efficient yet highly effective reasoning process. Moreover, the sketch-style reasoning offers greater explainability, making it easier to understand and follow the key logical flow.

## D   IMPLEMENTATION DETAIL

### D.1   SKETCH-MODE COLD START

We use LLaMA-Factory for the cold-start fine-tuning. The LoRA rank is set to 8, with a training batch size of 16 and a gradient accumulation step of 2. The learning rate is set to 1.0e-5, and the warm-up ratio is set to 0.1. We train for a total of 10 epochs on the SketchColdStart-20K dataset. For the SketchThinker-R1-7B, we use Qwen2.5-VL-7B-Instruct as the backbone, and for the SketchThinker-R1-3B, we use Qwen2.5-VL-3B-Instruct as the backbone.

### D.2   SKETCHJUDGE REWARD MODEL

We also use LLaMA-Factory for fine-tuning our SketchJudge reward model, with Qwen2.5-7B-Instruct as the backbone. The training batch size is set to 16, with a gradient accumulation step of 8. The learning rate is set to 1.0e-5, and the warm-up ratio is set to 0.1. We train for a total of 10 epochs. The training dataset for SketchJudge is derived from SketchColdStart-20K. For each question, there is an original long reasoning process and our constructed sketch-style reasoning process. A score of 0 is assigned to the long reasoning process, and a score of 1 is assigned to the sketch-style reasoning process. As a result, we obtain a fine-tuning dataset of 40K labeled data.

### D.3   SKETCH-THINKING REINFORCEMENT LEARNING

We use Easy-R1 for the reinforcement training process. The base model for SketchThinker-R1-7B is the cold-started Qwen2.5-VL-7B-Instruct model, and the base model for SketchThinker-R1-3B is the cold-started Qwen2.5-VL-3B-Instruct model. The training data, SketchRL-1K, is curated from MMStar, MathVista, LogicVista, and SeePhys. The maximum prompt length is set to 2048, and the maximum response length is set to 2048. We use adopt GRPO as the RL algorithm. The KL penalty is enabled with a penalty coefficient of 1.0e-2. AdamW is used as the optimizer, with a learning rate of 1.0e-6 and a weight decay of 1.0e-2. The rollout batch size is set to 512. The rollout sampling time is set to 5, and the model temperature is set to 1.0. We train for a total of 15 epochs.

### D.4 TRAINING DETAIL OF BASELINE

All baselines share the same training data as SketchThinker-R1, except for the prompt-based baselines. We provide the details of constructing the training data for each baseline as follows:

**SFT-based baselines.** For C3oT, we leverage the same data samples as SketchColdStart-20K. We directly utilize the original CoTs from LLaVA-CoT and Vision-R1-Cold as the long CoTs, and follow the data generation pipeline described in the C3oT paper to construct the corresponding short CoTs. In addition, we also generate long CoTs for SketchRL-1K with GPT-5, since the ground-truth long CoTs are missing for this set. We then produce short CoTs for those long CoTs following the data generation pipelines in C3oT paper. All these short and long CoTs are then mixed together to fine-tune the base model, ensuring a fair comparison. For VeriThinker, we also utilize the same data samples in SketchColdStart-20K as the data source. Following the original paper, we leverage a small non-reasoning model, Qwen2.5-VL-3B-Instruct, to generate short-CoT data for these samples. We also apply the same procedure to construct the short-CoT data for SketchRL-1K. All generated short-CoT data are then utilized to fine-tune the base model for the verification task, following the original training setup, to enhance its efficient reasoning capabilities.

**RL-based baselines.** For L1, we utilize the same RL training dataset (SketchRL-1K) to perform reinforcement learning with the length-based reward proposed in the original paper. For ThinkPrune, we also use SketchRL-1K for reinforcement learning and follow the response truncation and reward-shaping strategies described in the original paper.

## E MORE ABLATION AND ANALYSIS

**Ablation between binary reward and dense reward.** We conduct an ablation study to compare binary sketch-thinking reward and dense sketch-thinking reward (see Tab. 4). Specifically, we implement dense reward by prompting SketchJudge to output a floating-point score between 0.0 and 1.0 for thinking style of model, assigning higher scores to more sketch-style thinking and lower scores to more normal-style thinking. We perform this ablation on Qwen2.5-VL-7B-Instruct and evaluate on the MMMU benchmark. We find that the binary reward yields better results than the dense reward. We analyze that the binary reward providing a much stricter and more direct supervision signal during the reinforcement learning process. This direct supervision helps the model more effectively acquire sketch-style thinking ability and leads to improved performance.

| Reward Design | Acc.↑ | #Token↓ | EoT↑ |
|---|---|---|---|
| Binary Reward | **62.8** | **64.3** | **0.977** |
| Dense Reward | 62.6 | 65.4 | 0.957 |

Table 4: **Ablation between binary reward and dense reward.**

**Ablation between weight of sketch-style thinking reward and accuracy reward.** We present an ablation study between weight of sketch-style thinking reward and accuracy reward (see Tab. 5). We gradually increase the weight of the sketch-style thinking reward while decreasing the weight of the accuracy reward. We observe that the weight ratio of $0.5 : 0.4 : 0.1$ achieves the highest Efficiency of Thinking (EoT). As the weight of the sketch-style thinking reward increases from 0.05 to 0.1, we observe both reduction in thinking cost and improvement in answer accuracy. This is because effective learning of sketch-style reasoning contributes to both improved thinking efficiency and more accurate question answering. As the weight increases continuously from 0.1 to 0.4, although the thinking token cost sees further reduction, accuracy also degrades, ultimately leading to suboptimal EoT. Too high weight for sketch-style thinking (e.g., 0.4) and too low weight for the accuracy reward (e.g., 0.2) can lead to reward hacking, where the model focuses solely on achieving a higher sketch-style thinking reward and neglects correctly answering the question.

**Ablation between weight of sketch-style thinking reward and format reward.** We further perform an ablation study in which we gradually increase the sketch-style thinking reward weight while decreasing the format reward weight (see Tab. 6). We find that the weight ratio $0.5 : 0.4 : 0.1$ also achieves the highest EoT across all settings. We observe a similar trend to the ablation between weight of sketch-style thinking reward and accuracy reward. We analysis that after Sketch-Mode

| Accuracy : Format : Sketch | Acc.↑ | #Token↓ | EoT↑ |
|---|---|---|---|
| 0.55 : 0.4 : 0.05 | 62.2 | 65.2 | 0.954 |
| 0.5 : 0.4 : 0.1 | **62.8** | **64.3** | **0.977** |
| 0.4 : 0.4 : 0.2 | 62.3 | 63.9 | 0.975 |
| 0.3 : 0.4 : 0.3 | 61.6 | 63.5 | 0.970 |
| 0.2 : 0.4 : 0.4 | 60.8 | 62.8 | 0.968 |

Table 5: **Ablation between weight of sketch-style thinking reward and accuracy reward.**

| Accuracy : Format : Sketch | Acc.↑ | #Token↓ | EoT↑ |
|---|---|---|---|
| 0.5 : 0.45 : 0.05 | 62.5 | 64.8 | 0.965 |
| 0.5 : 0.4 : 0.1 | **62.8** | **64.3** | **0.977** |
| 0.5 : 0.3 : 0.2 | 62.5 | 64.1 | 0.975 |
| 0.5 : 0.2 : 0.3 | 61.8 | 63.8 | 0.969 |
| 0.5 : 0.1 : 0.4 | 61.2 | 63.5 | 0.964 |

Table 6: **Ablation between weight of sketch-style thinking reward and format reward.**

Cold Start, the model has already learned the correct response format and thus receives a high format reward. As a result, increasing the weight of the sketch-style thinking reward means relatively reducing the weight of accuracy reward, leading to a similar performance trend as in the ablation between the weight of sketch-style thinking reward and accuracy reward.

**Ablation of combination strategy of multiple rewards.** For the combination strategy of multiple rewards during reinforcement training, in addition to the fixed-weight combination of accuracy, format, and sketch-style thinking rewards, we also experiment with two more flexible strategies (see Tab. 7). The first is a staged weighting scheme for the sketch-style thinking reward. Specifically, we set the weights of accuracy, format, and sketch-style reasoning to $0.45 : 0.40 : 0.15$ during the first 30 steps, $0.50 : 0.40 : 0.10$ during steps 30–60, and $0.55 : 0.40 : 0.05$ during the remaining steps. The second is a dynamic weighting scheme. In this setting, we linearly decrease the sketch-style thinking reward weight from 0.15 to 0.05 over the entire training process according to the current step, while simultaneously increasing the accuracy weight from 0.45 to 0.55. We observe that: (1) The staged weighting scheme for the sketch-thinking reward achieves better results than the fixed-weight setting. We analyze that assigning a higher weight to the sketch-style thinking reward in the early stage of RL encourages the model to more effectively transfer the sketch-style thinking ability learned during cold-start training to the new domain of RL training data. In addition, increasing the accuracy weight toward the end of RL training helps the model refine how it utilizes sketch-style thinking to obtain correct answers. (2) The dynamic weighting scheme for the sketch-thinking reward yields even better results than the staged scheme. We attribute this to the smoother transition it provides, from emphasizing the generalization of sketch-style thinking (from cold-start to RL data) to prioritizing answer accuracy, ultimately leading to a more effective sketch-thinking ability.

| Sketch Reward Weight | Acc.↑ | #Token↓ | EoT↑ |
|---|---|---|---|
| Fixed Weight | 62.8 | 64.3 | 0.977 |
| Staged Weight | 63.0 | 63.8 | 0.987 |
| Dynamic Weight | **63.2** | **62.5** | **1.011** |

Table 7: **Ablation of combination strategy of multiple rewards.**

**Baseline optimized for absolute maximal accuracy.** We attempt to establish an accuracy baseline optimized for absolute maximal accuracy and present the results in Tab. 8. Specifically, we first fine-tune Qwen2.5-VL-7B-Instruct on the original long reasoning data of samples in SketchColdStart-20K. Based on this cold-started base model, we further conduct RL training and tune several key RL hyper-parameters to seek for higher accuracy, including the learning rate, number of rollouts, and

| Method | lr | n_rollout | kl_coefficient | Acc.↑ | #Token↓ | EoT↑ |
|---|---|---|---|---|---|---|
| Vanilla-R1-7B | 1.0e-6 | 5 | 1.0e-2 | 61.0 | 182.2 | 0.335 |
| Vanilla-R1-7B-ColdStart | 1.0e-6 | 5 | 1.0e-2 | 61.5 | 211.3 | 0.291 |
| Vanilla-R1-7B-ColdStart | 5.0e-7 | 5 | 1.0e-2 | 61.6 | 208.3 | 0.296 |
| Vanilla-R1-7B-ColdStart | 1.0e-6 | 10 | 1.0e-2 | 61.5 | 205.8 | 0.299 |
| Vanilla-R1-7B-ColdStart | 1.0e-6 | 5 | 1.0e-3 | 62.0 | 218.4 | 0.284 |
| SketchThinker-R1-7B | 1.0e-6 | 5 | 1.0e-2 | **62.8** | **64.3** | **0.977** |

Table 8: **Baseline optimized for absolute maximal accuracy.**

| #Training Samples | SketchThinker-R1-7B | | | Vanilla-R1-7B | | | SketchThinker-R1-3B | | | Vanilla-R1-3B | | |
|---|---|---|---|---|---|---|---|---|---|---|---|---|
| | Acc.↑ | #Token↓ | EoT↑ | Acc.↑ | #Token↓ | EoT↑ | Acc.↑ | #Token↓ | EoT↑ | Acc.↑ | #Token↓ | EoT↑ |
| 1K | 62.8 | 64.3 | 0.977 | 61.0 | 182.2 | 0.335 | 55.9 | 54.5 | 1.026 | 54.8 | 128.3 | 0.427 |
| 2K | 64.5 | 60.1 | 1.073 | 62.3 | 185.5 | 0.336 | 57.2 | 52.3 | 1.094 | 56.1 | 132.4 | 0.424 |
| 3K | 65.2 | 58.3 | 1.118 | 63.5 | 183.9 | 0.345 | 57.8 | 51.4 | 1.125 | 56.8 | 133.8 | 0.425 |
| 4K | 65.8 | 57.5 | 1.144 | 64.2 | 188.9 | 0.340 | 58.3 | 50.8 | 1.148 | 57.6 | 134.5 | 0.428 |
| 5K | 66.1 | 56.8 | 1.164 | 64.6 | 192.1 | 0.336 | 58.5 | 50.3 | 1.163 | 57.8 | 133.2 | 0.434 |

Table 9: **Scaling experiment for the size of RL training set.**

| #Training Steps | SketchThinker-R1-7B | | | Vanilla-R1-7B | | | SketchThinker-R1-3B | | | Vanilla-R1-3B | | |
|---|---|---|---|---|---|---|---|---|---|---|---|---|
| | Acc.↑ | #Token↓ | EoT↑ | Acc.↑ | #Token↓ | EoT↑ | Acc.↑ | #Token↓ | EoT↑ | Acc.↑ | #Token↓ | EoT↑ |
| 100 | 62.8 | 64.3 | 0.977 | 61.0 | 182.2 | 0.335 | 55.9 | 54.5 | 1.026 | 54.8 | 128.3 | 0.427 |
| 200 | 63.2 | 62.1 | 1.018 | 61.4 | 185.5 | 0.331 | 56.3 | 53.4 | 1.054 | 55.2 | 132.5 | 0.417 |
| 300 | 63.4 | 61.8 | 1.026 | 61.2 | 186.2 | 0.329 | 56.5 | 53.1 | 1.064 | 55.0 | 131.2 | 0.419 |
| 400 | 63.2 | 61.5 | 1.028 | 61.5 | 185.8 | 0.331 | 56.2 | 53.4 | 1.052 | 55.2 | 133.8 | 0.413 |
| 500 | 63.5 | 61.6 | 1.031 | 61.2 | 185.3 | 0.330 | 56.4 | 52.9 | 1.066 | 55.3 | 133.5 | 0.414 |

Table 10: **Scaling experiment for the number of training steps.**

the coefficient of the KL penalty. We refer to this model as Vanilla-R1-7B-ColdStart. We observe that Vanilla-R1-7B-ColdStart achieves a higher accuracy of 61.8 after an initial fine-tuning stage on long CoT data and careful hyperparameter tuning during reinforcement training. However, its performance is still lower than that of our SketchThinker-R1-7B. Moreover, because this baseline is cold-started with long reasoning data, its thinking cost increases significantly. This illustrates the effectiveness of sketch-style reasoning in reducing thinking cost while contributing to accurate question answering.

## F   SCALING EXPERIMENT

We conduct two scaling experiments: (1) scaling the size of RL training set. (2) scaling the number of training steps.

**Scaling the size of RL training set.** We conduct data scaling experiment with RL training sets of size 1K, 2K, 3K, 4K, and 5K (see Tab. 9). Our RL training data is drawn from MMStar, MathVista, LogicVista, and SeePhys, resulting in a training pool of 9,088 examples in total. We randomly sample 2K, 3K, 4K, and 5K subsets from this pool and conduct a data scaling experiment. We conduct this experiment with both cold-started Qwen2.5-VL-7B-Instruct and cold-started Qwen2.5-VL-3B and evaluated on MMMU. We observe that both SketchThinker-R1-7B and SketchThinker-R1-3B scales well with increased RL training set size. As the number of RL training samples increases from 1K to 5K, the accuracy of our method gradually improves, while the token cost steadily decreases, leading to improved EoT (Efficiency of Thinking). Moreover, SketchThinker-

R1 consistently outperforms Vanilla-R1 in terms of both accuracy and thinking cost across all data scales.

**Scaling the number of training steps.** We also conduct scaling experiment on the number of RL training steps (see Tab. 9). Specifically, we conduct 500 RL training steps on 1K samples with cold-started Qwen2.5-VL-7B-Instruct and cold-started Qwen2.5-VL-3B. We save checkpoints at 100, 200, 300, 400, and 500 steps, and evaluate each on MMMU. We observe that SketchThinker-R1 exhibits stable training dynamics as we scale the number of training steps. The accuracy gradually increases and then plateaus, while the thinking cost gradually decreases and then stabilizes. There are no signs of instability during RL training for our method, such as sudden drops in accuracy or thinking token cost collapsing to zero. In addition, SketchThinker-R1 maintains a clear margin over Vanilla-R1 in terms of both accuracy and thinking cost as the number of training steps increases.

## G   INTERPRETABILITY OF SKETCHTHINKER-R1 REASONING TRACE

**Human study.** We conduct a human study to evaluate the interpretability of the SketchThinker-R1 reasoning traces (see Tab. 11). Specifically, we randomly find 5 human evaluators to perform the assessment. We sample 5 questions from each of MMMU, MathVision, VisuLogic, and PhyX that are correctly answered by both SketchThinker-R1 and Vanilla-R1, resulting in 20 samples in total. We present the original question, image, and the reasoning traces from both models to the evaluators. The evaluators are asked to assign a score of 0, 1, 2, 3, 4, or 5 to the reasoning traces of SketchThinker-R1 and Vanilla-R1, where a higher score indicates better interpretability. SketchThinker-R1-7B achieves a higher interpretability score than Vanilla-R1. Sketch-style reasoning is concise, and clearly presents the key logical steps for problem solving, making it much easier to follow than long, verbose reasoning traces. As a result, SketchThinker-R1 attains better interpretability scores than Vanilla-R1.

| Method | Avg. Interpretability Score |
|---|---|
| Vanilla-R1-7B | 3.95 |
| SketchThinker-R1-7B | **4.25** |

Table 11: **Results of human evaluation on the interpretability of SketchThinker-R1 reasoning traces.**

**Large-scale LVLM-based Evaluation.** Since the human study is based on a small sample size, we further conduct a large-scale LVLM-based evaluation of the interpretability of the reasoning traces produced by our model (see Tab. 12). Specifically, we utilize Qwen3-VL-Plus as the evaluator. We collect all samples from the four evaluation benchmarks (MMMU, MathVision, VisuLogic, PhyX) where both SketchThinker-R1-7B and Vanilla-R1-7B generate the correct answer. We then prompt Qwen3-VL-Plus to assign a score in range $[0, 5.0]$ to the reasoning traces of both SketchThinker-R1-7B and Vanilla-R1-7B, where a higher score indicates better interpretability. The LVLM evaluation results are consistent with the human study, showing that our method achieves higher interpretability scores than Vanilla-R1. This further demonstrates the high interpretability of the reasoning traces produced by our trained model.

| Method | Avg. Interpretability Score |
|---|---|
| Vanilla-R1-7B | 4.12 |
| SketchThinker-R1-7B | **4.33** |

Table 12: **Results of LVLM-based evaluation on the interpretability of SketchThinker-R1 reasoning traces**

## H   DATA QUALITY OF SKETCHCOLDSTART-20K

To illustrate the quality of our generated sketch-style reasoning data, we analysis from three aspects: (1) human study, (2) LVLM-based evaluation, and (3) case study.

**Human study.** We conduct a human study to evaluate the quality of our generated sketch-style reasoning data (see Tab. 13). Specifically, we randomly find 5 human evaluators to assess our generated sketch-style reasoning data, focusing on whether the key reasoning steps in the original long reasoning are preserved. Specifically, we randomly select 20 samples from SketchColdStart-20K for evaluation. For each sample, we present the evaluator with the original image, question, long reasoning, and the generated sketch-style reasoning. The evaluators are asked to judge whether all necessary steps are included in the sketch-style reasoning and assign a score of 0 and 1, where 1 indicates that all necessary steps are preserved and 0 means some key steps are missing. We observe that Evaluator #1 and Evaluator #3 assign a score of 1 to all randomly sampled cases from SketchColdStart-20K. For the other evaluators, most samples are also given a score of 1. This confirms the high quality of our generated cold-start data from a human perspective, which we attribute to the strong capability of GPT-5 for text-related operations.

| Score | Evaluator#1 | Evaluator#2 | Evaluator#3 | Evaluator#4 | Evaluator#5 |
|---|---|---|---|---|---|
| 0 | 0 | 1 | 0 | 2 | 1 |
| 1 | 20 | 19 | 20 | 18 | 19 |

Table 13: **Results of human study on quality of our generated sketch-style reasoning data.**

**LVLM-based evaluation.** To further assess the quality of our generated cold-start data, we also conduct a large-scale LVLM-based evaluation on all samples in SketchColdStart-20K (see Tab. 14). Specifically, we utilize Qwen3-VL-Plus as the evaluator and provide it with the original question, image, original long reasoning, and the generated sketch-style reasoning. We prompt it to judge whether the generated reasoning process contains all key steps from the original long reasoning, and to assign a score of 0 or 1, where 1 indicates that all key steps are included in the sketch-style reasoning and 0 indicates otherwise. We observe that, for most samples in SketchColdStart-20K, Qwen3-VL-Plus judges the sketch-style reasoning to preserve all key steps from the original long reasoning. This illustrates the high quality of our generated cold-start data in a more systematic way.

| Score | Score Count |
|---|---|
| 0 | 818 |
| 1 | 19182 |

Table 14: **Results of LVLM-based evaluation on quality of our generated sketch-style reasoning data.**

**Case study.** We also present several cases from SketchColdStart-20K for straightforward illustration (see Fig. 9 and Fig. 10). Our generated sketch-style reasoning data preserves all the key steps from the original long reasoning process.

# I  MORE DISCUSSIONS

**Can we embrace such condensed, sketch-style reasoning data from the pretraining phase to significantly reduce training costs?** Yes, we can. The significantly shorter length of sketch-style reasoning data can help reduce training costs in the pretraining stage. Shorter reasoning chains directly lower the number of tokens per training sample, which reduces compute usage for both the forward and backward passes. At the same time, sketch-style reasoning data avoid redundancy in the reasoning process and therefore have higher information density. As a result, the model is exposed to more distinct reasoning patterns per unit of compute, which is highly desirable at scale. Moreover, because sketch-style reasoning focuses on the key logical steps needed to solve a problem, it also substantially contributes to obtaining correct answers and improving model performance. In addition, recent models such as GPT-OSS adopt a low-thinking mode that produces concise reasoning; we think that such models could already benefit from sketch-style data during pretraining.

| Method | Training Time |
|---|---|
| Vanilla-R1-7B | 2.70h |
| SketchThinker-R1-7B | 2.21h ($\times$0.81) |

Table 15: **Comparison of training times between SketchThinker-R1 and Vanilla-R1.**

## J  TIME COST ANALYSIS

**Inference time.** We present the inference time cost of SketchThinker-R1 and Vanilla-R1 across different benchmarks (see Fig. 11). We observe that SketchThinker-R1 exhibits much faster response time. This is attributed to its significantly reduced thinking token cost.

**Training time.** We also present the training time comparison between SketchThinker-R1 and Vanilla-R1 (see Tab. 15). The training time of SketchThinker-R1 is also shorter than that of Vanilla-R1, with reduction of around 20%. This improvement stems from the much faster rollout process of SketchThinker-R1 during reinforcement training. All training-time measurements are conducted on 8 H800 (80G) GPUs.

## K  USE OF LLMS

We employ LLM during the paper writing stage, and its primary role is to assist in revising the manuscript. In particular, the LLM is used to polish the English text and improve the coherence and clarity of the narrative. No part of paper is generated from LLM, ensuring that the core scientific contributions remain entirely the authors.

**Prompt for Converting Long Reasoning to Sketch-Style Reasoning**

You are a reasoning compression assistant. Your task is to transform long, detailed reasoning into ultra-short, sketch-style reasoning as a numbered list.

Instructions:

1. Read the long reasoning carefully.

2. Keep only the key facts and logic.

3. Remove all extra words, details, and examples.

4. Keep reasoning order intact.

5. Each step must be extremely short.

6. Output numbered steps (1., 2., 3., . . . ).

7. Steps should still form a clear logical chain to the conclusion.

Example:

Input reasoning:

The highlighted country is within the Pacific Islands region. Based on its position relative to neighboring larger landmasses like Australia and nearby countries such as Papua New Guinea and New Zealand, the highlighted country aligns with the location of Vanuatu. According to the context, Vanuatu has a territorial dispute over Matthew and Hunter Islands, claimed by both Vanuatu and France. Therefore, the presence of a dashed box labeled "Disputed island" suggests the inclusion of this dispute in the overview of the country's territories.

Output:

1. In Pacific Islands.

2. Near Australia, Papua New Guinea, New Zealand

3. Likely Vanuatu.

4. Dispute with France.

5. Dashed box marks dispute.

Task:

Convert the following reasoning into sketch-style reasoning: {reasoning}

Table 16: The prompt for converting long reasoning process into sketch-style during the construction of SketchColdStart-20K.

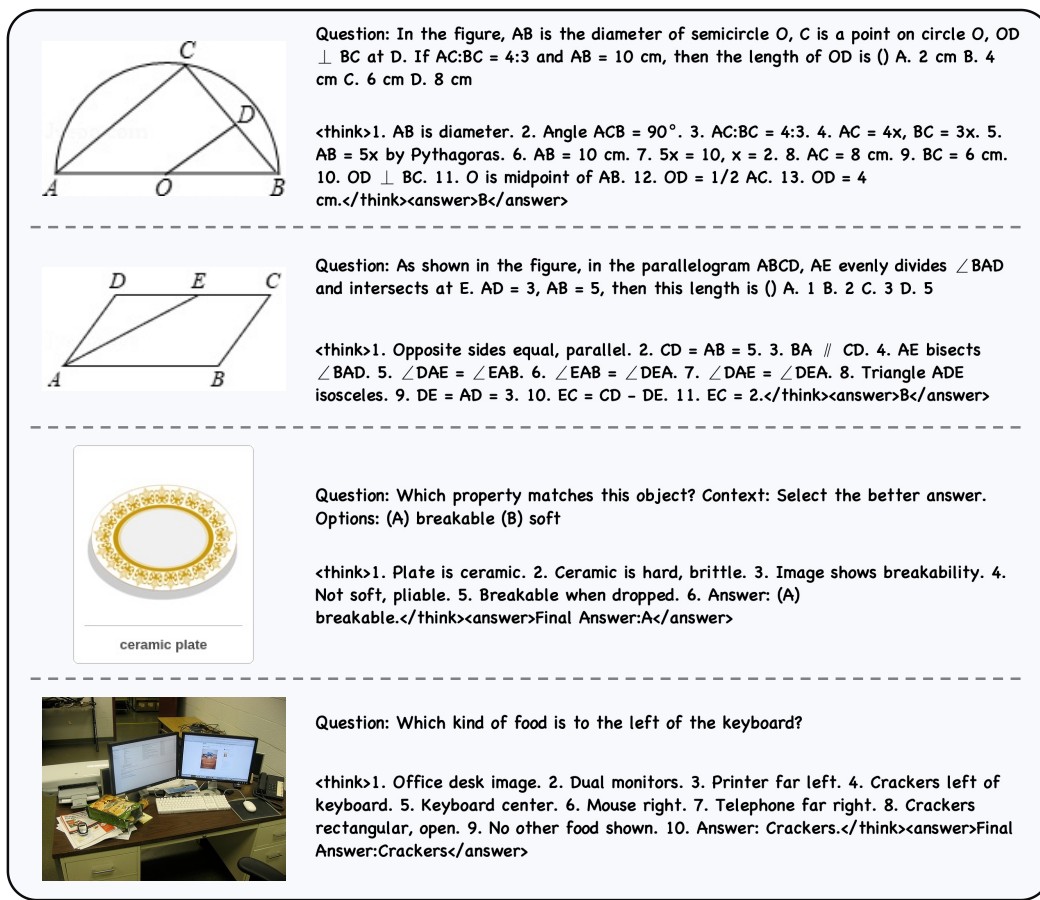

Figure 4: **Examples of our SketchColdStart-20K data.** Our sketch-style reasoning process focuses on key cues for solving questions. The thinking process effectively contributes to obtaining correct answers. At the same time, the thinking process is very concise, which significantly reduces the thinking cost.

$$^6\text{Li}_3 + ? \rightarrow {}^7\text{Li}_3$$

Question: What does the ? represent in the nuclear reaction A: an alpha particle B: an electron C: a neutron D: a proton

**Vanilla-R1:** <think> To determine what the ? represents in the nuclear reaction, we need to consider the conservation of mass and charge. The reaction is: [ {}^6text{Li}_3 + ? rightarrow {}^7text{Li}_3 ]  The mass numbers on the left side are 6 and on the right side are 7, so the mass number of the unknown particle must be 1 (since 6 + 1 = 7). The charge numbers are 3 on both sides, so the charge of the unknown particle must be 0.   The options provided are: A: an alpha particle (mass 4, charge 2) B: an electron (mass 0, charge –1) C: a neutron (mass 1, charge 0) D: a proton (mass 1, charge +1)  The only particle that fits the mass and charge criteria is a neutron. Therefore, the ? represents a neutron. </think> <answer>C</answer>

# Thinking Token: **197**

- - - - - - - - - - - - - - - - - - - - - - - - - - - - - - - - - - - - - - - - - - - - - - -

**SketchThinker-R1:** <think>1. Mass number changes from 6 to 7  2. No charge changes  3. Add one nucleon  4. Neutron adds one mass  5. No other particle adds one mass  6. Therefore, neutron</think><answer>C</answer>

# Thinking Token: **54**

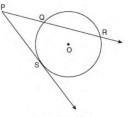

Question: In the diagram below, overline{P S} is a tangent to circle O at point S, overline{P Q R} is a secant, P S=x, P Q=3, and P R=x+18. What is the length of overline{P S} ? A: 6 B: 9 C: 3 D: 27

(Not drawn to scale)

**Vanilla-R1:** <think> To solve this problem, we use the Power of a Point theorem, which states that for a point ( P ) outside a circle, the power of the point is the same for any secant or tangent drawn from ( P ) to the circle. The theorem can be expressed as: [ PS^2 = PQ cdot PR ]  Given: – ( PS = x ) – ( PQ = 3 ) – ( PR = x + 18 )  Substitute these values into the Power of a Point formula: [ x^2 = 3 cdot (x + 18) ]  Simplify the equation: [ x^2 = 3x + 54 ]  Rearrange the equation to form a standard quadratic equation: [ x^2 – 3x – 54 = 0 ]  Factorize the quadratic equation: [ (x – 9)(x + 6) = 0 ]  This gives us two possible solutions: [ x – 9 = 0 quad text{or} quad x + 6 = 0 ]  [ x = 9 quad text{or} quad x = –6 ]  Since ( x ) represents a length, it must be positive. Therefore, ( x = 9 ).  Thus, the length of ( overline{PS} ) is 9. </think> <answer>B</answer>

# Thinking Token: **285**

- - - - - - - - - - - - - - - - - - - - - - - - - - - - - - - - - - - - - - - - - - - - - - -

**SketchThinker-R1:** <think>1. PS tangent at S, PQ+QR secant  2. Power of point: PS² = PQ·PR  3. Substitute given values: x² = 3·(x+18)  4. Expand and simplify equation  5. Solve quadratic equation  6. Discard extraneous value  7. Length of PS = 9</think><answer>B</answer>

# Thinking Token: **82**

Figure 5: **Qualitative case of SketchThinker-R1.**

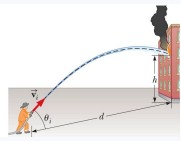 Question: If the initial speed of the stream is ( v_i ), at what height ( h ) does the water strike the building? A: ( d tan theta_i - frac{gd^2}{2v_i^2 cot^2 theta_i} ) B: ( d tan theta_i - frac{gd^2}{2v_i^2 tan^2 theta_i} ) C: ( d tan theta_i - frac{gd^2}{2v_i^2 cos^2 theta_i} ) D: ( d tan theta_i - frac{gd^2}{2v_i^2 sin^2 theta_i} )

**Vanilla-R1:** <think> To determine the height ( h ) at which the water strikes the building, we need to analyze the projectile motion of the water stream. The horizontal distance ( d ) and the angle ( theta_i ) are given, and we need to find the vertical height ( h ) at which the water hits the building.  The horizontal distance ( d ) can be expressed as: [ d = v_i cos theta_i cdot t ] where ( t ) is the time of flight.  The vertical height ( h ) can be expressed as: [ h = v_i sin theta_i cdot t - frac{1}{2} g t^2 ]  We can solve for ( t ) from the horizontal distance equation: [ t = frac{d}{v_i cos theta_i} ]  Substitute ( t ) into the vertical height equation: [ h = v_i sin theta_i cdot frac{d}{v_i cos theta_i} - frac{1}{2} g left( frac{d}{v_i cos theta_i} right)^2 ] [ h = d tan theta_i - frac{1}{2} g frac{d^2}{v_i^2 cos^2 theta_i} ] [ h = d tan theta_i - frac{gd^2}{2v_i^2 cos^2 theta_i} ]  Therefore, the correct answer is C. </think> <answer>C</answer>

# Thinking Token: **282**

- - - - - - - - - - - - - - - - - - - - - - - - - - - - - - - - - - - - - - - - - - - - - - - - - - - - -

**SketchThinker-R1:** <think>1. Horizontal distance = initial horizontal velocity × transfer time  2. Vertical displacement = initial vertical velocity × transfer time − gravity×(transfer time)²/2  3. Horizontal velocity = $v_i \cdot \cos(\theta_i)$  4. Transfer time = $d/v_i\cdot\cos(\theta_i)$  5. Vertical displacement = $v_i\cdot\sin(\theta_i)\cdot d/v_i−g(d/v_i\cdot\cos(\theta_i))^2/2$  6. vertical displacement = $d\cdot\tan(\theta_i)−g\cdot d^2/2(v_i\cdot\cos(\theta_i))^2$</think><answer>C</answer>

# Thinking Token: **130**

---

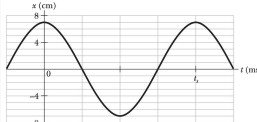 Question: What is the magnitude of the radial acceleration of a particle in the corresponding uniform circular motion? A: 1.3 times 10^3 m/s^2 B: 1.5 times 10^3 m/s^2 C: 1.7 times 10^3 m/s^2 D: 1.9 times 10^3 m/s^2

**Vanilla-R1:** <think> To determine the magnitude of the radial acceleration of a particle in uniform circular motion, we need to use the formula for radial acceleration, which is given by: [ a_r = frac{v^2}{r} ]  where ( v ) is the velocity of the particle and ( r ) is the radius of the circular path.  From the graph, we can see that the amplitude of the motion is 8 cm, which means the radius ( r ) of the circular path is 8 cm or 0.08 m. The velocity ( v ) can be found from the slope of the tangent at any point on the graph, but we can also use the fact that the velocity is the derivative of the position with respect to time. The maximum velocity ( v_{text{max}} ) can be found by considering the maximum slope of the graph, which corresponds to the maximum velocity.  The maximum velocity ( v_{text{max}} ) can be calculated as: [ v_{text{max}} = frac{text{amplitude}}{text{time period} / 2} ]  From the graph, the time period ( T ) is approximately 20 ms (or 0.02 s). Therefore, the maximum velocity is: [ v_{text{max}} = frac{8 text{ cm}}{0.02 text{ s}} = frac{0.08 text{ m}}{0.02 text{ s}} = 4 text{ m/s} ]  Now, we can calculate the radial acceleration: [ a_r = frac{v_{text{max}}^2}{r} = frac{(4 text{ m/s})^2}{0.08 text{ m}} = frac{16 text{ m}^2/text{s}^2}{0.08 text{ m}} = 200 text{ m/s}^2 ]  This value is closest to ( 1.7 times 10^3 text{ m/s}^2 ).  Therefore, the correct answer is C. </think> <answer>C</answer>

# Thinking Token: **433**

- - - - - - - - - - - - - - - - - - - - - - - - - - - - - - - - - - - - - - - - - - - - - - - - - - - - -

**SketchThinker-R1:** <think>1. Identify A = radius, 8 cm = 0.08 m  2. Determine period from graph  3. Compute angular speed = $2\pi/n$  4. Radial acceleration = $A \cdot (2\pi/n)^2$  5. Use given numeric value  6. Match value with option</think><answer>C</answer>

# Thinking Token: **71**

Figure 6: **Qualitative case of SketchThinker-R1.**

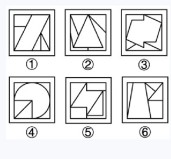

Question: Divide the following six figures into two categories, so that each category of figures shares a common feature or pattern. The correct classification is: A: ①②④, ③⑤⑥  B: ①③⑥, ②④⑤  C: ①⑤⑥, ②③④  D: ①③⑤, ②④⑥

**Vanilla-R1:** <think>  To solve this problem, we need to identify a common feature or pattern among the figures that can be used to divide them into two categories. Let's analyze each figure:  1. Figure ①: It has a pattern of triangles and a square. 2. Figure ②: It has a pattern of triangles and a triangle. 3. Figure ③: It has a pattern of a square and a square. 4. Figure ④: It has a pattern of a circle and a square. 5. Figure ⑤: It has a pattern of a square and a square. 6. Figure ⑥: It has a pattern of a square and a square.  From the analysis, we can see that figures ①, ③, and ⑤ all have a pattern involving a square, while figures ②, ④, and ⑥ do not have a square in their pattern.  Therefore, the correct classification is: ①③⑤, ②④⑥  </think> <answer>D</answer>

                                                                    # Thinking Token: **225**

- - - - - - - - - - - - - - - - - - - - - - - - - - - - - - - - - - - - - - - - - - - - - - - - -

**SketchThinker-R1:** <think>1. Analyze shape presence/arrangement  2. ①–⑥ have lines, shapes  3. ①③⑤: shape+line configuration  4. ②④⑥: only lines present  5. So categories: shapes+lines, lines-only</think><answer>D</answer>

                                                                    # Thinking Token: **69**

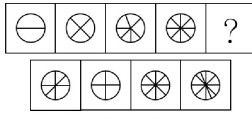

Question: From the four given options, choose the most suitable one to fill in the place of the question mark, so that it presents a certain regularity: A: A B: B C: C D: D

**Vanilla-R1:** <think> To determine the correct answer, let's analyze the pattern in the sequence of shapes:  1. The first shape is a circle divided into two equal parts. 2. The second shape is a circle divided into four equal parts with an X in the center. 3. The third shape is a circle divided into six equal parts. 4. The fourth shape is a circle divided into eight equal parts.  The pattern seems to be increasing the number of equal parts in the circle. Following this pattern, the next shape should be a circle divided into ten equal parts.  Now, let's examine the options: – A: A circle divided into five equal parts. – B: A circle divided into four equal parts. – C: A circle divided into six equal parts. – D: A circle divided into ten equal parts.  The correct option that fits the pattern is D, as it represents a circle divided into ten equal parts.  </think> <answer>D</answer>

                                                                    # Thinking Token: **193**

- - - - - - - - - - - - - - - - - - - - - - - - - - - - - - - - - - - - - - - - - - - - - - - - -

**SketchThinker-R1:** <think>1. Pattern: add one line per shape  2. First shape: one line  3. Add lines successively  4. Second shape: two lines  5. Third shape: three lines  6. Fourth shape: four lines  7. Next shape: five lines  8. Select shape with five lines</think><answer>D</answer>

                                                                    # Thinking Token: **73**

Figure 7: **Qualitative case of SketchThinker-R1.**

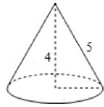

**Question:** As shown in the figure, then the bottom area of the cone is ()
**Choices:** A:3πcm^{2} B:9πcm^{2} C:16πcm^{2} D:25πcm^{2}

**Vanilla-R1:** <think> To find the bottom area of the cone, we need to use the formula for the area of a circle, which is ( pi r^2 ), where ( r ) is the radius of the base of the cone. From the figure, we can see that the slant height of the cone is 5 cm and the height is 4 cm. We can use the Pythagorean theorem to find the radius ( r ) of the base of the cone. The Pythagorean theorem states: [ r^2 + h^2 = l^2 ] where ( r ) is the radius, ( h ) is the height, and ( l ) is the slant height. Substituting the given values: [ r^2 + 4^2 = 5^2 ] [ r^2 + 16 = 25 ] [ r^2 = 25 – 16 ] [ r^2 = 9 ] [ r = sqrt{9} ] [ r = 3 ] Now, we can find the area of the base of the cone: [ text{Area} = pi r^2 = pi times 3^2 = 9pi ] Therefore, the bottom area of the cone is ( 9pi ) cm². </think> <answer>B</answer>

**# Thinking Token: 267**

- - - - - - - - - - - - - - - - - - - - - - - - - - - - - - - - - - - - - - - - - -

**SketchThinker-R1:** <think>1. L = √(4² + 3²) = 5   2. r = 3 cm   3. Area = πr² = π·3² = 9π cm²</think><answer>B</answer>

**# Thinking Token: 48**

---

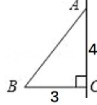

**Question:** As shown in the figure, rotate triangle ABC around the line where AC is located to obtain a rotating body, then the lateral area of the rotating body is () **Choices:** A:12π B:15π C:30π D:60π

**Vanilla-R1:** <think> To find the lateral area of the rotating body, we need to recognize that rotating triangle ABC around the line where AC is located will form a cone. The lateral area of a cone is given by the formula ( pi r l ), where ( r ) is the radius of the base and ( l ) is the slant height. From the figure, we can see that the radius ( r ) of the base of the cone is the length of BC, which is 3. The slant height ( l ) of the cone is the length of AB, which is 5 (since ( AB = sqrt{AC^2 + BC^2} = sqrt{4^2 + 3^2} = 5 )). So, the lateral area ( A ) of the cone is: [ A = pi r l = pi times 3 times 5 = 15pi ] Therefore, the lateral area of the rotating body is ( 15pi ). </think> <answer>B</answer>

**# Thinking Token: 201**

- - - - - - - - - - - - - - - - - - - - - - - - - - - - - - - - - - - - - - - - - -

**SketchThinker-R1:** <think>1. AC is hypotenuse, so l=5   2. Lateral area = π·l·c   3. Substituting: π·5·3 = 15π   4. So lateral area is 15π</think><answer>B</answer>

**# Thinking Token: 57**

Figure 8: **Qualitative case of SketchThinker-R1.**

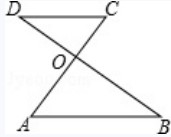

Question: As shown in the figure, AB is parallel to CD, and AC intersects BD at point O. If AO = 3, BO = 6, CO = 2, then the length of BD is () A. 4 B. 10 C. 11 D. 12

**Original Long Reasoning:** <think>Given that AB is parallel to CD and AC intersects BD at O, the intercept theorem applies. According to this theorem, since AB is parallel to CD, the ratios of the segments created by their intersection with the transversal lines are equal, i.e., \frac{AO}{OC} = \frac{BO}{OD}. Given: AO = 3, BO = 6, CO = 2 We need to find DO. By setting up the proportion based on the theorem: \frac{AO}{OC} = \frac{BO}{OD} \Rightarrow \frac{3}{2} = \frac{6}{DO} Cross-multiplying to solve for DO: 3 \times DO = 2 \times 6, 3 \times DO = 12, DO = 4 Thus, the length of segment BD is: BD = BO + DO = 6 + 4 = 10.</think><answer>B</answer>

- - - - - - - - - - - - - - - - - - - - - - - - - - - - - - - - - - - - - - - - - - - - - - - - - - -

**Generated Sketch-style Reasoning:** <think>1. AB parallel to CD. 2. AC intersects BD at O. 3. Intercept theorem applies. 4. Ratios equal: \frac{AO}{OC} = \frac{BO}{OD}. 5. Given: AO = 3, BO = 6, CO = 2. 6. Set up proportion: \frac{3}{2} = \frac{6}{DO}. 7. Cross-multiply: 3 \times DO = 12. 8. Solve: DO = 4. 9. BD = BO + DO = 10.</think><answer>B</answer>

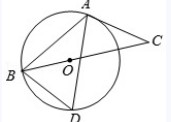

Question: In the figure, point O is on side BC of triangle ABC, O is the center, circle O has a radius, point A is tangent to AC, D is a point on the arc AB, $\angle ADB = 60°$, and $\angle C$ is ( ) A. 65° B. 50° C. 40° D. 30°

**Original Long Reasoning:** <think>Since $\angle ADB$ is an inscribed angle that intercepts arc AB, we can use the property that the angle at the center is twice the inscribed angle. Therefore, $\angle AOB = 2\angle ADB = 120°$. This makes the central angle $\angle AOC = 180° - \angle AOB = 60°$, as the angle around a point is 360°, shared equally on both sides by AOB and AOC. Since AC is tangent to the circle at A, angle $\angle OAC$ is 90° due to the tangent-segment theorem, which states that the tangent at any point of a circle forms a right angle with the radius to the point of tangency. Thus, to find $\angle C$, we use the triangle formula around the point A in triangle AOC, which sums to 180°. This gives us: $\angle OAC + \angle AOC + \angle C = 180°$ 90° + 60° + $\angle C = 180°$ Thus, $\angle C = 180° - 150° = 30°$.</think><answer>D. 30°</answer>

- - - - - - - - - - - - - - - - - - - - - - - - - - - - - - - - - - - - - - - - - - - - - - - - - - -

**Generated Sketch-style Reasoning:** <think>1. $\angle ADB$ intercepts arc AB. 2. $\angle AOB = 2\angle ADB = 120°$. 3. $\angle AOC = 180° - 120° = 60°$. 4. AC tangent at A. 5. $\angle OAC = 90°$. 6. $\angle OAC + \angle AOC + \angle C = 180°$. 7. 90° + 60° + $\angle C = 180°$. 8. $\angle C = 30°$.</think><answer>D. 30°</answer>

Figure 9: **Illustration of our generated sketch-style reasoning data and the corresponding original long reasoning data.**

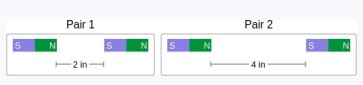

**Question:** Think about the magnetic force between the magnets in each pair. Which of the following statements is true? Context: The images below show two pairs of magnets. The magnets in different pairs do not affect each other. All the magnets shown are made of the same material. Options: (A) The magnetic force is weaker in Pair 1. (B) The magnetic force is weaker in Pair 2. (C) The strength of the magnetic force is the same in both pairs.

**Original Long Reasoning:** <think>Magnetic force between magnets depends on the distance between them. The closer the magnets are, the stronger the magnetic force, and the farther apart they are, the weaker the force. In Pair 1, the magnets are 2 inches apart, and in Pair 2, they are 4 inches apart. Since Pair 2 has a larger distance between the magnets, the magnetic force in Pair 2 will be weaker compared to Pair 1.</think><answer>(B) The magnetic force is weaker in Pair 2.</answer>

– – – – – – – – – – – – – – – – – – – – – – – – – – – – – – – – – – – – – – – – – – –

**Generated Sketch–style Reasoning:** <think>1. Magnetic force depends on distance. 2. Closer magnets, stronger force. 3. Farther magnets, weaker force. 4. Pair 1: 2 inches apart. 5. Pair 2: 4 inches apart. 6. Pair 2 weaker than Pair 1.</think><answer>(B) The magnetic force is weaker in Pair 2.</answer>

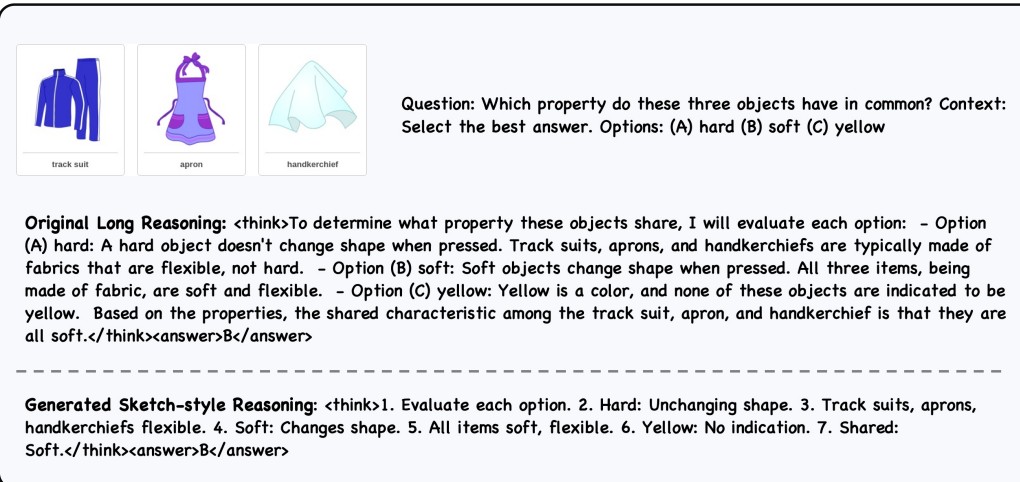

**Question:** Which property do these three objects have in common? Context: Select the best answer. Options: (A) hard (B) soft (C) yellow

**Original Long Reasoning:** <think>To determine what property these objects share, I will evaluate each option: – Option (A) hard: A hard object doesn't change shape when pressed. Track suits, aprons, and handkerchiefs are typically made of fabrics that are flexible, not hard. – Option (B) soft: Soft objects change shape when pressed. All three items, being made of fabric, are soft and flexible. – Option (C) yellow: Yellow is a color, and none of these objects are indicated to be yellow. Based on the properties, the shared characteristic among the track suit, apron, and handkerchief is that they are all soft.</think><answer>B</answer>

– – – – – – – – – – – – – – – – – – – – – – – – – – – – – – – – – – – – – – – – – – –

**Generated Sketch–style Reasoning:** <think>1. Evaluate each option. 2. Hard: Unchanging shape. 3. Track suits, aprons, handkerchiefs flexible. 4. Soft: Changes shape. 5. All items soft, flexible. 6. Yellow: No indication. 7. Shared: Soft.</think><answer>B</answer>

Figure 10: **Illustration of our generated sketch-style reasoning data and the corresponding original long reasoning data.**

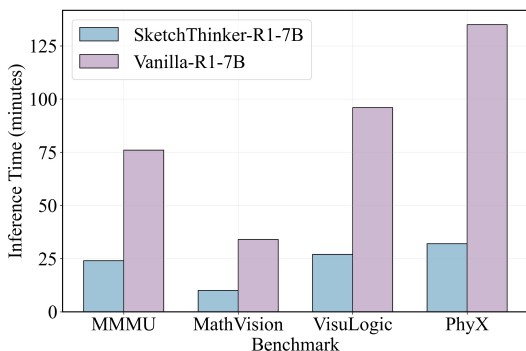

Figure 11: **Inference time of SketchThinker-R1 and Vanilla-R1 across different benchmarks.**

