# OpenReview forum: "SketchThinker-R1: Towards Efficient Sketch-Style Reasoning in Large Multimodal Models"
_ICLR.cc/2026/Conference — ICLR 2026 Poster_

### Official Review · Reviewer_5YsX · 2025-10-30

**Soundness:** 3
**Presentation:** 3
**Contribution:** 3
**Rating:** 6
**Confidence:** 4

**Summary:**

This paper presents SketchThinker-R1, a reinforcement learning framework designed to induce sketch-style reasoning in large multimodal models—defined as concise, goal-directed reasoning that retains essential logical steps while eliminating redundancy. The method is composed of three stages: (1) Sketch-Mode Cold Start, in which long-form reasoning traces from LLaVA-CoT-100K and Vision-R1-cold are systematically distilled into sketch-style sequences via GPT-5, followed by supervised fine-tuning to instill initial reasoning condensation capability; (2) SketchJudge Reward Model training, where an open-source LLM (Qwen2.5-7B-Instruct) is fine-tuned on paired reasoning samples (normal vs. sketch-style) to learn a reliable binary score for reasoning style, with no reliance on absolute length thresholds; and (3) Sketch-Thinking Reinforcement Learning, wherein the cold-started model is further optimized via GRPO, with rewards composed of accuracy (50%), format adherence (40%), and SketchJudge style score (10%). Evaluated across four standard multimodal benchmarks—MMMU, MathVision, VisuLogic, and PhyX—the method achieves an average reduction of 64.3% in reasoning token cost compared to the Vanilla-R1 baseline, with no degradation in answer accuracy and consistently superior efficiency.

**Strengths:**

1. This paper introduces a novel and comprehensive three-stage framework, SketchThinker-R1, which moves beyond superficial reasoning length constraints by directly fostering an "intrinsic sketch-style thinking" within large multimodal models. This approach represents a significant paradigm shift from external compression to internal cognitive efficiency.

2. The development and integration of the specialized SketchJudge reward model are a key strength. This model's ability to accurately evaluate "reasoning style" rather than merely length provides a precise and stable signal for reinforcement learning, enabling the model to learn context-aware reasoning condensation.

3. The framework consistently achieves substantial reductions in reasoning token cost (over 64% on average) across diverse benchmarks (MMMU, MathVision, VisuLogic, PhyX) while maintaining or even improving answer accuracy. This demonstrates a robust solution to the critical challenge of high computational inference costs in advanced multimodal models.

4. This paper benefits from clear writing, well-structured sections, and effective use of visual aids. Specifically, Figure 1 (quantitative results overview) and Figure 3 (qualitative reasoning examples) are particularly effective in conveying the core idea and the method's impact, making complex concepts accessible to the reader.

**Weaknesses:**

1. This paper relies on a powerful closed-source large language model for generating concise "sketch-style reasoning" data. It remains uncertain whether these generated data consistently encapsulate all necessary reasoning steps, which could potentially impact the interpretability of the subsequent black-box reasoning model and challenge the paper's assumption of mimicking human thought.

2. This paper claims to enhance efficiency without compromising accuracy; however, its accuracy baseline for comparison does not represent a model optimized for "absolute maximal accuracy." If a model, trained solely on the original long reasoning data (with consistent samples but without prior condensation), could achieve higher accuracy, then SketchThinker-R1 might be achieving its efficiency gains at a suboptimal accuracy ceiling, suggesting an unexplored trade-off between accuracy and efficiency.

**Questions:**

Considering the findings and conclusions presented in this paper, does this imply that we could embrace such condensed, sketch-style reasoning data from the pretraining phase to significantly reduce training costs?

---

> ### Author Response · Authors · 2025-11-21
> **Response to Reviewer 5YsX (Part 1/3)**
>
> Dear Reviewer 5YsX,
>
> We would like to express our sincere gratitude for your valuable feedbacks and the time you devoted to reviewing our work. Thank you for your insightful comments and positive recommendation. We are encouraged that you view our work as a transition from imposing superficial reasoning-length constraints toward building an intrinsic sketch-style efficient reasoning ability. We provide point-by-point responses to address your concerns below.
>
> ---
>
> **Comment (1)**: *"This paper relies on a powerful closed-source large language model for generating concise "sketch-style reasoning" data. It remains uncertain whether these generated data consistently encapsulate all necessary reasoning steps, which could potentially impact the interpretability of the subsequent black-box reasoning model and challenge the paper's assumption of mimicking human thought."*
>
> **Response (1)**: Thanks for your valuable comment and pointing this out. To illustrate the quality of our generated sketch-style reasoning data, we will response from three aspects: (1) human study, (2) LVLM-based evaluation, and (3) case study.
>
> - **Human study.** For the human study, we randomly find 5 human evaluators to assess our generated sketch-style reasoning data, focusing on whether the key reasoning steps in the original long reasoning are preserved. Specifically, we randomly select 20 samples from SketchColdStart-20K for evaluation. For each sample, we present the evaluator with the original image, question, long reasoning, and the generated sketch-style reasoning. The evaluators are asked to judge whether all necessary steps are included in the sketch-style reasoning and assign a score of 0 and 1, where 1 indicates that all necessary steps are preserved and 0 means some key steps are missing. The human study results are as follows, where we present the score count of each human evaluator:
>
>     | **Score** | **Evaluator#1** | **Evaluator#2** | **Evaluator#3** | **Evaluator#4** | **Evaluator#5** |
>     |-------|:-------------:|:-------------:|:-------------:|:-------------:|:-------------:|
>     | **0**     | 0           | 1           | 0           | 2           | 1           |
>     | **1**     | 20          | 19          | 20          | 18          | 19          |
>
>     We observe that Evaluator #1 and Evaluator #3 assign a score of 1 to all randomly sampled cases from SketchColdStart-20K. For the other evaluators, most samples are also given a score of 1. This confirms the high quality of our generated cold-start data from a human perspective, which we attribute to the strong capability of GPT-5 for text-related operations.
>
> - **LVLM-based evaluation.** To further assess the quality of our generated cold-start data, we also conduct a large-scale LVLM-based evaluation on all samples in SketchColdStart-20K. Specifically, we utilize Qwen3-VL-Plus as the evaluator and provide it with the original question, image, original long reasoning, and the generated sketch-style reasoning. We prompt it to judge whether the generated reasoning process contains all key steps from the original long reasoning, and to assign a score of 0 or 1, where 1 indicates that all key steps are included in the sketch-style reasoning and 0 indicates otherwise. The evaluation results are as follows:
>
>     | **Score** | **Score Count** |
>     |-------|:-------------:|
>     | **0**     | 818         |
>     | **1**     | 19182       |
>
>     We observe that, for most samples in SketchColdStart-20K, Qwen3-VL-Plus judges the sketch-style reasoning to preserve all key steps from the original long reasoning. This illustrates the high quality of our generated cold-start data in a more systematic way.

---

> ### Author Response · Authors · 2025-11-21
> **Response to Reviewer 5YsX (Part 2/3)**
>
> **[ Continue the response to above Comment (1)]**
>
> - **Case study.** We also present several cases from SketchColdStart-20K for straightforward illustration. We show the original long reasoning and the generated sketch-style reasoning here, and provide the full samples, including the images and questions, in Fig. 9 and Fig. 10 in our latest upload of revised paper.
>
>     - **Case1**
>
>         - **Original Long Reasoning**: \<think>Given that AB is parallel to CD and AC intersects BD at O, the intercept theorem applies. According to this theorem, since AB is parallel to CD, the ratios of the segments created by their intersection with the transversal lines are equal, i.e.,  \frac{AO}{OC} =  \frac{BO}{OD}.  Given: AO = 3, BO = 6, CO = 2  We need to find DO. By setting up the proportion based on the theorem: \frac{AO}{OC} =  \frac{BO}{OD} \Rightarrow  \frac{3}{2} =  \frac{6}{DO} Cross-multiplying to solve for DO: 3 \times DO = 2 \times 6, 3 \times DO = 12, DO = 4 Thus, the length of segment BD is: BD = BO + DO = 6 + 4 = 10.\</think>\<answer>B\</answer>
>
>         - **Generated Sketch-style Reasoning**: \<think>1. AB parallel to CD. 2. AC intersects BD at O. 3. Intercept theorem applies. 4. Ratios equal: \frac{AO}{OC} = \frac{BO}{OD}. 5. Given: AO = 3, BO = 6, CO = 2. 6. Set up proportion: \frac{3}{2} = \frac{6}{DO}. 7. Cross-multiply: 3 \times DO = 12. 8. Solve: DO = 4. 9. BD = BO + DO = 10.\</think>\<answer>B\</answer>
>
>     - **Case2**
>
>         - **Original Long Reasoning**: \<think>Since ∠ADB is an inscribed angle that intercepts arc AB, we can use the property that the angle at the center is twice the inscribed angle. Therefore, ∠AOB = 2∠ADB = 120°. This makes the central angle ∠AOC = 180° - ∠AOB = 60°, as the angle around a point is 360°, shared equally on both sides by AOB and AOC.  Since AC is tangent to the circle at A, angle ∠OAC is 90° due to the tangent-segment theorem, which states that the tangent at any point of a circle forms a right angle with the radius to the point of tangency. Thus, to find ∠C, we use the triangle formula around the point A in triangle AOC, which sums to 180°. This gives us:  ∠OAC + ∠AOC + ∠C = 180°  90° + 60° + ∠C = 180°  Thus, ∠C = 180° - 150° = 30°.\</think>\<answer>D. 30°\</answer>
>
>         - **Generated Sketch-style Reasoning**: \<think>1. ∠ADB intercepts arc AB. 2. ∠AOB = 2∠ADB = 120°. 3. ∠AOC = 180° - 120° = 60°. 4. AC tangent at A. 5. ∠OAC = 90°. 6. ∠OAC + ∠AOC + ∠C = 180°. 7. 90° + 60° + ∠C = 180°. 8. ∠C = 30°.\</think>\<answer>D. 30°\</answer>
>
>     - **Case3**
>
>         - **Original Long Reasoning**: \<think>Magnetic force between magnets depends on the distance between them. The closer the magnets are, the stronger the magnetic force, and the farther apart they are, the weaker the force. In Pair 1, the magnets are 2 inches apart, and in Pair 2, they are 4 inches apart. Since Pair 2 has a larger distance between the magnets, the magnetic force in Pair 2 will be weaker compared to Pair 1.\</think>\<answer>(B) The magnetic force is weaker in Pair 2.\</answer>
>
>         - **Generated Sketch-style Reasoning**: \<think>1. Magnetic force depends on distance. 2. Closer magnets, stronger force. 3. Farther magnets, weaker force. 4. Pair 1: 2 inches apart. 5. Pair 2: 4 inches apart. 6. Pair 2 weaker than Pair 1.\</think>\<answer>(B) The magnetic force is weaker in Pair 2.\</answer>
>
>     - **Case4**
>
>         - **Original Long Reasoning**: \<think>To determine what property these objects share, I will evaluate each option:  - Option (A) hard: A hard object doesn't change shape when pressed. Track suits, aprons, and handkerchiefs are typically made of fabrics that are flexible, not hard.  - Option (B) soft: Soft objects change shape when pressed. All three items, being made of fabric, are soft and flexible.  - Option (C) yellow: Yellow is a color, and none of these objects are indicated to be yellow.  Based on the properties, the shared characteristic among the track suit, apron, and handkerchief is that they are all soft.\</think>\<answer>B\</answer>
>
>         - **Generated Sketch-style Reasoning**: \<think>1. Evaluate each option. 2. Hard: Unchanging shape. 3. Track suits, aprons, handkerchiefs flexible. 4. Soft: Changes shape. 5. All items soft, flexible. 6. Yellow: No indication. 7. Shared: Soft.\</think>\<answer>B\</answer>
>
>     We observe that, across various domains such as mathematical, scientific, and general QA, our generated sketch-style reasoning effectively encapsulates all key logical steps present in the original long reasoning traces.
>
> We have added the those discussions in Appendix H of our latest upload of revised paper (pages 18-19, lines 968–1011).

---

> ### Author Response · Authors · 2025-11-21
> **Response to Reviewer 5YsX (Part 3/3)**
>
> **Comment (2)**: *"This paper claims to enhance efficiency without compromising accuracy; however, its accuracy baseline for comparison does not represent a model optimized for "absolute maximal accuracy." If a model, trained solely on the original long reasoning data (with consistent samples but without prior condensation), could achieve higher accuracy, then SketchThinker-R1 might be achieving its efficiency gains at a suboptimal accuracy ceiling, suggesting an unexplored trade-off between accuracy and efficiency."*
>
> **Response (2)**: Thanks for your valuable feedback. Following your comment, we attempt to establish an accuracy baseline optimized for "absolute maximal accuracy". Specifically, we first fine-tune Qwen2.5-VL-7B-Instruct on the original long reasoning data of samples in SketchColdStart-20K. Based on this cold-started base model, we further conduct RL training and tune several key RL hyper-parameters to seek for higher accuracy, including the learning rate, number of rollouts, and the coefficient of the KL penalty. We refer to this model as Vanilla-R1-7B-ColdStart. The results are as follows:
>
> | **Method**                  | **lr**     | **n_rollout** | **kl_coefficient** | **Acc. $\uparrow$**  | **#Token $\downarrow$** | **EoT $\uparrow$**    |
> |-------------------------|:--------:|:-----------:|:----------------:|:-------:|:--------:|:--------:|
> | Vanilla-R1-7B           | 1.0e-6 | 5         | 1.0e-2         | 61.0  | 182.2  | 0.335  |
> | Vanilla-R1-7B-ColdStart | 1.0e-6 | 5         | 1.0e-2         | 61.5  | 211.3  | 0.291  |
> | Vanilla-R1-7B-ColdStart | 5.0e-7 | 5         | 1.0e-2         | 61.6  | 208.3  | 0.296  |
> | Vanilla-R1-7B-ColdStart | 1.0e-6 | 10        | 1.0e-2         | 61.5  | 205.8  | 0.299  |
> | Vanilla-R1-7B-ColdStart | 1.0e-6 | 5         | 1.0e-3         | 62.0  | 218.4  | 0.284  |
> | SketchThinker-R1-7B     | 1.0e-6 | 5         | 1.0e-2         | **62.8**  | **64.3**   | **0.977**  |
>
> We observe that Vanilla-R1-7B-ColdStart achieves a higher accuracy of 61.8 after an initial fine-tuning stage on long CoT data and careful hyperparameter tuning during reinforcement training. However, its performance is still lower than that of our SketchThinker-R1-7B. Moreover, because this baseline is cold-started with long reasoning data, its thinking cost increases significantly. This illustrates the effectiveness of sketch-style reasoning in reducing thinking cost while contributing to accurate question answering.
>
> We have added the those discussions in Appendix E of our latest upload of revised paper (pages 16-17, lines 860–903).
>
> **Comment (3)**: *"Considering the findings and conclusions presented in this paper, does this imply that we could embrace such condensed, sketch-style reasoning data from the pretraining phase to significantly reduce training costs?"*
>
> **Response (3)**: Thanks for your inspiring comment. Yes, we agree. The significantly shorter length of sketch-style reasoning data can help reduce training costs in the pretraining stage. Shorter reasoning chains directly lower the number of tokens per training sample, which reduces compute usage for both the forward and backward passes. At the same time, sketch-style reasoning data avoid redundancy in the reasoning process and therefore have higher information density. As a result, the model is exposed to more distinct reasoning patterns per unit of compute, which is highly desirable at scale. Moreover, because sketch-style reasoning focuses on the key logical steps needed to solve a problem, it also substantially contributes to obtaining correct answers and improving model performance. In addition, recent models such as GPT-OSS adopt a low-thinking mode that produces concise reasoning; we think that such models could already benefit from sketch-style data during pretraining.
>
> We have added the those discussions in Appendix I of our latest upload of revised paper (pages 19, lines 1014–1025).
>
> ---
>
> We hope that our responses have addressed your concerns. If you have additional comments or concerns, please let us know and we will be more than happy to answer.
>
> Best,
>
> Authors

---

> > ### Comment · Reviewer_5YsX · 2025-11-26
> >
> > Thanks for the rebuttal. I believe this work is a meaningful contribution toward developing a reinforcement learning framework that induces sketch-based reasoning in large multimodal models. I will maintain my current positive rating.

---

> > > ### Author Response · Authors · 2025-11-27
> > >
> > > Dear Reviewer 5YsX,
> > >
> > > We sincerely appreciate the time and effort you invested in reviewing our manuscript. We greatly appreciate your positive recommendation! If you have additional comments or concerns, please let us know and we will be more than happy to answer.
> > >
> > > Best,
> > >
> > > Authors

---

### Official Review · Reviewer_ag5r · 2025-11-01

**Soundness:** 3
**Presentation:** 3
**Contribution:** 3
**Rating:** 6
**Confidence:** 4

**Summary:**

This paper proposes SketchThinker-R1, a three-stage framework to train large multimodal models to produce “sketch-style” reasoning chains: first a cold-start stage where long form reasoning traces are converted into concise sketches and the model is fine-tuned on them; second a SketchJudge reward model is trained to explicitly score reasoning traces by favoring the sketch style; third a reinforcement-learning (RL) stage (using GRPO) encourages the model to produce sketch-style reasoning under the reward model’s supervision. Their motivation is that current large multimodal reasoning models use long chain-of-thought traces, which incur high token cost and increased latency, while human “sketching” uses fewer but more salient steps — thus they aim to reduce reasoning token cost without sacrificing accuracy. They build a dataset “SketchColdStart-20K” by converting 20K long-form reasoning traces (10 K from each of two multimodal reasoning datasets) into sketch‐style reasoning via a strong LLM (GPT-5) and then fine-tune the base model. They then sample a small RL set of 1 K from four domains (MMStar, MathVista, LogicVista, SeePhys) for the RL stage.

**Strengths:**

The motivation is good. Reducing token number is reasonable for multimodal tasks thinking.

The three‐stage pipeline (cold‐start conversion of reasoning style, reward model, RL) is reasonably well‐designed and coherently described. The reward model is reasonable.

Experiments consistently show good results spanning different domains (visual reasoning, logic, physics) while showing substantial reductions in token cost while preserving good accuracy.

**Weaknesses:**

1. Suggest an experiment on ratio of the sketch reward. What if increasing the sketch coefficient and lower the format coefficient?

2. It is unclear if every baseline shares the same training data or just vanilla r1. Good to include more details.

3. How much time (GPU hour) in practice could be saved?

**Questions:**

See weakness.

---

> ### Author Response · Authors · 2025-11-21
> **Response to Reviewer ag5r (Part 1/2)**
>
> Dear Reviewer ag5r,
>
> We thank you for your valuable comments and the time you spent reviewing our work! Your professional feedback provides valuable guidance for making our paper more comprehensive and competitive. We address the raised concerns and questions below.
>
> ---
>
> **Comment (1)**: *"Suggest an experiment on ratio of the sketch reward. What if increasing the sketch coefficient and lower the format coefficient?"*
>
> **Response (1)**: Thank you for the valuable advice. We conduct two ablation studies for weight of sketch-style thinking reward: (1) Ablation between weight of sketch-style thinking reward and accuracy reward. (2) Ablation between weight of sketch-style thinking reward and format reward. Specifically, we conduct experiments with cold-started Qwen2.5-VL-7B-Instruct and evaluate on MMMU.
>
> **Ablation between weight of sketch-style thinking reward and accuracy reward**
>
> First, we gradually increase the weight of the sketch-style thinking reward while decreasing the weight of the accuracy reward. The results are shown below:
>
> | **Accuracy : Format : Sketch** | **Acc. $\uparrow$**  | **#Token $\downarrow$** | **EoT $\uparrow$**    |
> |----------------------------|:-------:|:--------:|:--------:|
> | 0.55 : 0.4 : 0.05          | 62.2  | 65.2   | 0.954  |
> | 0.5 : 0.4 : 0.1            | **62.8**  | 64.3   | **0.977**  |
> | 0.4 : 0.4 : 0.2            | 62.3  | 63.9   | 0.975  |
> | 0.3 : 0.4 : 0.3            | 61.6  | 63.5   | 0.970  |
> | 0.2 : 0.4 : 0.4            | 60.8  | **62.8**   | 0.968  |
>
> We observe that the weight ratio of 0.5 : 0.4 : 0.1 achieves the highest Efficiency of Thinking (EoT). As the weight of the sketch-style thinking reward increases from 0.05 to 0.1, we observe both reduction in thinking cost and improvement in answer accuracy. This is because effective learning of sketch-style reasoning contributes to both improved thinking efficiency and more accurate question answering. As the weight increases continuously from 0.1 to 0.4, although the thinking token cost sees further reduction, accuracy also degrades, ultimately leading to suboptimal EoT. Too high weight for sketch-style thinking (e.g., 0.4) and too low weight for the accuracy reward (e.g., 0.2) can lead to reward hacking, where the model focuses solely on achieving a higher sketch-style thinking reward and neglects correctly answering the question.
>
> **Ablation between weight of sketch-style thinking reward and format reward**
>
> We further perform an ablation study in which we gradually increase the sketch-style thinking reward weight while decreasing the format reward weight. The results are listed as follows:
>
> | **Accuracy : Format : Sketch** | **Acc. $\uparrow$**  | **#Token $\downarrow$** | **EoT $\uparrow$**    |
> |----------------------------|:-------:|:--------:|:--------:|
> | 0.5 : 0.45 : 0.05          | 62.5  | 64.8   | 0.965  |
> | 0.5 : 0.4 : 0.1            | **62.8**  | 64.3   | **0.977**  |
> | 0.5 : 0.3 : 0.2            | 62.5  | 64.1   | 0.975  |
> | 0.5 : 0.2 : 0.3            | 61.8  | 63.8   | 0.969  |
> | 0.5 : 0.1 : 0.4            | 61.2  | **63.5**   | 0.964  |
>
> We find that the weight ratio 0.5:0.4:0.1 also achieves the highest EoT across all settings. We observe a similar trend to the ablation between weight of sketch-style thinking reward and accuracy reward. We analysis that after Sketch-Mode Cold Start, the model has already learned the correct response format and thus receives a high format reward. As a result, increasing the weight of the sketch-style thinking reward means relatively reducing the weight of accuracy reward, leading to a similar performance trend as in the ablation between the weight of sketch-style thinking reward and accuracy reward.
>
> We have added the those discussions in Appendix E of our latest upload of revised paper (pages 15–16, lines 794–834).

---

> ### Author Response · Authors · 2025-11-21
> **Response to Reviewer ag5r (Part 2/2)**
>
> **Comment (2)**: *"It is unclear if every baseline shares the same training data or just vanilla r1. Good to include more details."*
>
> **Response (2)**: Thank you very much for your insightful question and constructive suggestions. All baselines share the same training data as SketchThinker-R1, except for the prompt-based baselines. We provide the details of constructing the training data for each baseline as follows:
>
> - **SFT-based baselines**
>
>     - For C3oT, we leverage the same data samples as SketchColdStart-20K. We directly utilize the original CoTs from LLaVA-CoT and Vision-R1-Cold as the long CoTs, and follow the data generation pipeline described in the C3oT paper to construct the corresponding short CoTs. In addition, we also generate long CoTs for SketchRL-1K with GPT-5, since the ground-truth long CoTs are missing for this set. We then produce short CoTs for those long CoTs following the data generation pipelines in C3oT paper. All these short and long CoTs are then mixed together to fine-tune the base model, ensuring a fair comparison.
>
>     - For VeriThinker, we also utilize the same data samples in SketchColdStart-20K as the data source. Following the original paper, we leverage a small non-reasoning model, Qwen2.5-VL-3B-Instruct, to generate short-CoT data for these samples. We also apply the same procedure to construct the short-CoT data for SketchRL-1K. All generated short-CoT data are then utilized to fine-tune the base model for the verification task, following the original training setup, to enhance its efficient reasoning capabilities.
>
> - **RL-based baselines**
>
>     - For L1, we utilize the same RL training dataset (SketchRL-1K) to perform reinforcement learning with the length-based reward proposed in the original paper.
>
>     - For ThinkPrune, we also use SketchRL-1K for reinforcement learning and follow the response truncation and reward-shaping strategies described in the original paper.
>
> We have added the those details in Appendix D.4 of our latest upload of revised paper (pages 15, lines 756–775).
>
> **Comment (3)**: *"How much time (GPU hour) in practice could be saved?"*
>
> **Response (3)**: Thanks for your valuable question. We observe clear reductions in both inference time and training time when comparing SketchThinker-R1 to Vanilla-R1. The inference time comparisons are shown below:
>
> | **Method**              | **MMMU**     | **MathVision** | **VisuLogic**            | **PhyX**                 |
> |---------------------|:----------:|:------------:|:----------------------:|:----------------------:|
> | Vanilla-R1-7B       | 1.26h               | 0.56h      | 1.60h                | 2.25h                |
> | SketchThinker-R1-7B | 0.40h ($\times$0.31) | 0.16h ($\times$0.28) | 0.45h ($\times$0.28) | 0.53h ($\times$0.23) |
>
> We observe that SketchThinker-R1 yields approximately 70% reduction in inference time compared with Vanilla-R1 across our evaluation benchmarks. This improvement is primarily due to significantly shorter reasoning traces of SketchThinker-R1, which greatly enhance its inference efficiency. All time-cost measurements are conducted on a single H800 (80G) GPU.
>
> For training time of RL, the comparison is listed as follows:
>
> | **Method**              | **Training Time** |
> |---------------------|:---------------:|
> | Vanilla-R1-7B       | 2.70h         |
> | SketchThinker-R1-7B | 2.21h ($\times$0.81) |
>
> The training time of SketchThinker-R1 is also shorter than that of Vanilla-R1, with reduction of around 20%. This improvement stems from the much faster rollout process of SketchThinker-R1 during reinforcement training. All training-time measurements are conducted on 8 H800 (80G) GPUs.
>
> We have added the those discussions in Appendix J of our latest upload of revised paper (pages 20, lines 1038–1043).
>
> ---
>
> We hope that our responses have addressed your concerns. If you have additional comments or concerns, please let us know and we will be more than happy to answer.
>
> Best,
>
> Authors

---

### Official Review · Reviewer_FKy6 · 2025-11-03

**Soundness:** 2
**Presentation:** 4
**Contribution:** 3
**Rating:** 6
**Confidence:** 4

**Summary:**

The paper introduces SketchThinker-R1, an RL framework that trains multimodal LM to elicient concise reasoning instead of verbose ones, while keeping the same level of accuracy. The method contains three steps. The first step is use a strong LLM to converte existing long COT reasoning traces into short ones with numbered steps, and then do cold-start SFT. The second step is to train a SketchJudge reward model that can classify if a reasoning trace is "sketch style". The third step is do GRPO with the verifiable reward, format reward, and the sketchjudge reward.

The authors experiment this method on MMMU, MathVision, VisuLogic, PhyX, and show that this method can reduce reasoning tokens by >64% while keeping the accuracy.

**Strengths:**

1. The method gives good efficiency gains on multimodal reasoning, with 64% token reduction, while keeping the accuracy.

2. The method is simple and make sense. It shows that by carefully curate the SFT cold start data, and a new reward for sketch-style, it is possible to achieve similar reasoning process while greatly reduce the token count.

3. The ablations are well designed and provide good analysis.

**Weaknesses:**

1. The major weakness is that the scale of the experiments are very small. It only contains 1K prompts training set and 150 steps. It is questionable whether this method is scalable and generalizable on more tasks. What will happen if a bigger training set is used, and more FLOPs trained? Will it be continually improving, or this method is more unstable compared with vanilla R1? The paper would be stronger if include more scaling experiments.

2. More analysis on the reward models would make the paper stronger. Now the method uses a 0/1 reward on the "sketch" style. What if we use a dense reward? Also, it has a weight of 0.1. What will happen if we use other weights? Any reward hacking, or competing with other rewards observed during experiments?

**Questions:**

1. About the reward model SketchJudge. What alternatives have you tried? Like for example, a dense reward. Why using a binary reward in the end?

2. How is the weight 0.5, 0.4, 0.1 determined? What will happen if using other weights. Does the SketchJudge hurts accuracy if the weight is too big? Is there other method to combine these multiple objectives?

3. Have you run a human study on the interpretablity of the reasoning traces of the models you trained? Are the reasoning traces concise and reasonable, or they are just shorter but with many unreadable stuff?

---

> ### Author Response · Authors · 2025-11-21
> **Response to Reviewer FKy6 (Part 1/5)**
>
> Dear Reviewer FKy6,
>
> We sincerely thank you for your insightful comments and the time you dedicated to reviewing our work. Your expert feedback has been invaluable in guiding us towards refining our paper and making it more comprehensive and competitive. We greatly appreciate your support and constructive suggestions. We address the raised concerns and questions below.
>
> ---
>
> **Comment (1)**: *"The major weakness is that the scale of the experiments are very small. It only contains 1K prompts training set and 150 steps. It is questionable whether this method is scalable and generalizable on more tasks. What will happen if a bigger training set is used, and more FLOPs trained? Will it be continually improving, or this method is more unstable compared with vanilla R1? The paper would be stronger if include more scaling experiments."*
>
> **Response (1)**: Thank you very much for your constructive feedback. Following your suggestions, we conduct two scaling experiments: (1) scaling the size of RL training set. (2) scaling the number of training steps.
>
> - **Scaling the size of RL training set.** For the size of the RL training set, we extend it up to a maximum of 5K samples. Our RL training data is drawn from MMStar, MathVista, LogicVista, and SeePhys, resulting in a training pool of 9,088 examples in total. We randomly sample 2K, 3K, 4K, and 5K subsets from this pool and conduct a data scaling experiment. We conduct this experiment with both cold-started Qwen2.5-VL-7B-Instruct and cold-started Qwen2.5-VL-3B and evaluated on MMMU. The results are as follows:
>
>     | **#Training Samples** |  |     **SketchThinker-R1-7B**   |        |  |   **Vanilla-R1-7B**     |        |  |    **SketchThinker-R1-3B**    |        |  |    **Vanilla-R1-3B**    |        |
>     |-------------------|:---------------------:|:--------:|:--------:|:---------------:|:--------:|:--------:|:---------------------:|:--------:|:--------:|:---------------:|:--------:|:--------:|
>     |                   | **Acc. $\uparrow$**                | **#Token $\downarrow$** | **EoT $\uparrow$**    | **Acc. $\uparrow$**          | **#Token $\downarrow$** | **EoT $\uparrow$**    | **Acc. $\uparrow$**                | **#Token $\downarrow$** | **EoT $\uparrow$**    | **Acc. $\uparrow$**          | **#Token $\downarrow$** | **EoT $\uparrow$**    |
>     | 1K                | 62.8                | 64.3   | 0.977  | 61.0          | 182.2  | 0.335  | 55.9                | 54.5   | 1.026  | 54.8          | 128.3  | 0.427  |
>     | 2K                | 64.5                | 60.1   | 1.073  | 62.3          | 185.5  | 0.336  | 57.2                | 52.3   | 1.094  | 56.1          | 132.4  | 0.424  |
>     | 3K                | 65.2                | 58.3   | 1.118  | 63.5          | 183.9  | 0.345  | 57.8                | 51.4   | 1.125  | 56.8          | 133.8  | 0.425  |
>     | 4K                | 65.8                | 57.5   | 1.144  | 64.2          | 188.9  | 0.340  | 58.3                | 50.8   | 1.148  | 57.6          | 134.5  | 0.428  |
>     | 5K                | 66.1                | 56.8   | 1.164  | 64.6          | 192.1  | 0.336  | 58.5                | 50.3   | 1.163  | 57.8          | 133.2  | 0.434  |
>
>     We observe that both SketchThinker-R1-7B and SketchThinker-R1-3B scales well with increased RL training set size. As the number of RL training samples increases from 1K to 5K, the accuracy of our method gradually improves, while the token cost steadily decreases, leading to improved EoT (Efficiency of Thinking). Moreover, SketchThinker-R1 consistently outperforms Vanilla-R1 in terms of both accuracy and thinking cost across all data scales. Thank you again for your comments, which help us further improve our paper.

---

> ### Author Response · Authors · 2025-11-21
> **Response to Reviewer FKy6 (Part 2/5)**
>
> **[ Continue the response to above Comment (1)]**
>
> - **Scaling the number of training steps.** For scaling the number of training steps, we conduct 500 RL training steps on 1K samples with cold-started Qwen2.5-VL-7B-Instruct and cold-started Qwen2.5-VL-3B. We save checkpoints at 100, 200, 300, 400, and 500 steps, and evaluate each on MMMU. The results are as follows:
>
>     | **#Training Steps** |  | **SketchThinker-R1-7B**       |        |  |  **Vanilla-R1-7B**      |        |  |  **SketchThinker-R1-3B**      |        |  |  **Vanilla-R1-3B**      |        |
>     |--------|:---------------------:|:--------:|:--------:|:---------------:|:--------:|:--------:|:---------------------:|:--------:|:--------:|:---------------:|:--------:|:--------:|
>     |        | **Acc. $\uparrow$**                | **#Token $\downarrow$** | **EoT $\uparrow$**    | **Acc. $\uparrow$**          | **#Token $\downarrow$** | **EoT $\uparrow$**    | **Acc. $\uparrow$**                | **#Token $\downarrow$** | **EoT $\uparrow$**    | **Acc. $\uparrow$**          | **#Token $\downarrow$** | **EoT $\uparrow$**    |
>     | 100    | 62.8                | 64.3   | 0.977  | 61.0          | 182.2  | 0.335  | 55.9                | 54.5   | 1.026  | 54.8          | 128.3  | 0.427  |
>     | 200    | 63.2                | 62.1   | 1.018  | 61.4          | 185.5  | 0.331  | 56.3                | 53.4   | 1.054  | 55.2          | 132.5  | 0.417  |
>     | 300    | 63.4                | 61.8   | 1.026  | 61.2          | 186.2  | 0.329  | 56.5                | 53.1   | 1.064  | 55.0          | 131.2  | 0.419  |
>     | 400    | 63.2                | 61.5   | 1.028  | 61.5          | 185.8  | 0.331  | 56.2                | 53.4   | 1.052  | 55.2          | 133.8  | 0.413  |
>     | 500    | 63.5                | 61.6   | 1.031  | 61.2          | 185.3  | 0.330  | 56.4                | 52.9   | 1.066  | 55.3          | 133.5  | 0.414  |
>
>     We observe that SketchThinker-R1 exhibits stable training dynamics as we scale the number of training steps. The accuracy gradually increases and then plateaus, while the thinking cost gradually decreases and then stabilizes. There are no signs of instability during RL training for our method, such as sudden drops in accuracy or thinking token cost collapsing to zero. In addition, SketchThinker-R1 maintains a clear margin over Vanilla-R1 in terms of both accuracy and thinking cost as the number of training steps increases.
>
> We have added the those discussions in Appendix F of our latest upload of revised paper (pages 17–18, lines 906–928).

---

> ### Author Response · Authors · 2025-11-21
> **Response to Reviewer FKy6 (Part 3/5)**
>
> **Comment (2)**: *"More analysis on the reward models would make the paper stronger. Now the method uses a 0/1 reward on the "sketch" style. What if we use a dense reward? Also, it has a weight of 0.1. What will happen if we use other weights? Any reward hacking, or competing with other rewards observed during experiments?"*
>
> **Response (2)**:  Thanks for your thoughtful feedback and insightful comment. Following your comments, we conduct a more detailed analysis of the SketchJudge reward model design.
>
> **Ablation between binary reward and dense reward**
>
> We conduct an ablation study to compare binary sketch-thinking reward and dense sketch-thinking reward. Specifically, we implement dense reward by prompting SketchJudge to output a floating-point score between 0.0 and 1.0 for thinking style of model, assigning higher scores to more sketch-style thinking and lower scores to more normal-style thinking. We perform this ablation on cold-started Qwen2.5-VL-7B-Instruct and evaluate on the MMMU benchmark. The results are as follows:
>
> | **Reward Design** | **Acc. $\uparrow$**  | **#Token $\downarrow$** | **EoT $\uparrow$**   |
> |---------------|:-------:|:--------:|:-------:|
> | Binary Reward | **62.8**  | **64.3**   | **0.977** |
> | Dense Reward  | 62.6  | 65.4   | 0.957 |
>
> We find that the binary reward yields better results than the dense reward. We analyze that the binary reward providing a much stricter and more direct supervision signal during the reinforcement learning process. This direct supervision helps the model more effectively acquire sketch-style thinking ability and leads to improved performance.
>
> We conduct two ablation studies for weight of sketch-style thinking reward: (1) Ablation between weight of sketch-style thinking reward and accuracy reward. (2) Ablation between weight of sketch-style thinking reward and format reward. Specifically, we conduct experiments with cold-started Qwen2.5-VL-7B-Instruct and evaluate on MMMU.
>
> **Ablation between weight of sketch-style thinking reward and accuracy reward**
>
> First, we gradually increase the weight of the sketch-style thinking reward while decreasing the weight of the accuracy reward. The results are shown below:
>
> | **Accuracy : Format : Sketch** | **Acc. $\uparrow$**  | **#Token $\downarrow$** | **EoT $\uparrow$**    |
> |----------------------------|:-------:|:--------:|:--------:|
> | 0.55 : 0.4 : 0.05          | 62.2  | 65.2   | 0.954  |
> | 0.5 : 0.4 : 0.1            | **62.8**  | 64.3   | **0.977**  |
> | 0.4 : 0.4 : 0.2            | 62.3  | 63.9   | 0.975  |
> | 0.3 : 0.4 : 0.3            | 61.6  | 63.5   | 0.970  |
> | 0.2 : 0.4 : 0.4            | 60.8  | **62.8**   | 0.968  |
>
> We observe that the weight ratio of 0.5 : 0.4 : 0.1 achieves the highest Efficiency of Thinking (EoT). As the weight of the sketch-style thinking reward increases from 0.05 to 0.1, we observe both reduction in thinking cost and improvement in answer accuracy. This is because effective learning of sketch-style reasoning contributes to both improved thinking efficiency and more accurate question answering. As the weight increases continuously from 0.1 to 0.4, although the thinking token cost sees further reduction, accuracy also degrades, ultimately leading to suboptimal EoT. Too high weight for sketch-style thinking (e.g., 0.4) and too low weight for the accuracy reward (e.g., 0.2) can lead to reward hacking, where the model focuses solely on achieving a higher sketch-style thinking reward and neglects correctly answering the question.
>
> **Ablation between weight of sketch-style thinking reward and format reward**
>
> We further perform an ablation study in which we gradually increase the sketch-style thinking reward weight while decreasing the format reward weight. The results are listed as follows:
>
> | **Accuracy : Format : Sketch** | **Acc. $\uparrow$**  | **#Token $\downarrow$** | **EoT $\uparrow$**    |
> |----------------------------|:-------:|:--------:|:--------:|
> | 0.5 : 0.45 : 0.05          | 62.5  | 64.8   | 0.965  |
> | 0.5 : 0.4 : 0.1            | **62.8**  | 64.3   | **0.977**  |
> | 0.5 : 0.3 : 0.2            | 62.5  | 64.1   | 0.975  |
> | 0.5 : 0.2 : 0.3            | 61.8  | 63.8   | 0.969  |
> | 0.5 : 0.1 : 0.4            | 61.2  | **63.5**   | 0.964  |
>
> We find that the weight ratio 0.5:0.4:0.1 also achieves the highest EoT across all settings. We observe a similar trend to the ablation between weight of sketch-style thinking reward and accuracy reward. We analysis that after Sketch-Mode Cold Start, the model has already learned the correct response format and thus receives a high format reward. As a result, increasing the weight of the sketch-style thinking reward means relatively reducing the weight of accuracy reward, leading to a similar performance trend as in the ablation between the weight of sketch-style thinking reward and accuracy reward.
>
> We have added the those discussions in Appendix E of our latest upload of revised paper (pages 15–16, lines 779–834).

---

> ### Author Response · Authors · 2025-11-21
> **Response to Reviewer FKy6 (Part 4/5)**
>
> **Comment (3)**: *"About the reward model SketchJudge. What alternatives have you tried? Like for example, a dense reward. Why using a binary reward in the end?"*
>
> **Response (3)**: Thanks for your valuable question. We primarily study the base model utilized to build SketchJudge reward model, and observe that a stronger model, which provides more accurate supervision signal for model thinking style during reinforcement training process, can facilitate better sketch-style thinking ability (see Tab. 3(d)). For the ablation between the binary sketch-thinking reward and the dense reward, please refer to **Response (2)**.
>
> **Comment (4)**: *"How is the weight 0.5, 0.4, 0.1 determined? What will happen if using other weights. Does the SketchJudge hurts accuracy if the weight is too big? Is there other method to combine these multiple objectives?"*
>
> **Response (4)**: Thank you very much for your insightful questions and valuable suggestions. For ablation study on the weight of the sketch-style thinking reward, please refer to **Response (2)** for a detailed analysis.
>
> For the combination of multiple objectives, in addition to the fixed-weight combination of accuracy, format, and sketch-style thinking rewards, we also experiment with two more flexible strategies. The first is a staged weighting scheme for the sketch-style thinking reward. Specifically, we set the weights of accuracy, format, and sketch-style reasoning to 0.45 : 0.40 : 0.15 during the first 30 steps, 0.50 : 0.40 : 0.10 during steps 30–60, and 0.55 : 0.40 : 0.05 during the remaining steps. The second is a dynamic weighting scheme. In this setting, we linearly decrease the sketch-style thinking reward weight from 0.15 to 0.05 over the entire training process according to the current step, while simultaneously increasing the accuracy weight from 0.45 to 0.55. The ablation results are as follows:
>
> | **Sketch Reward Weight** | **Acc. $\uparrow$**  | **#Token $\downarrow$** | **EoT $\uparrow$**    |
> |----------------------|:-------:|:--------:|:--------:|
> | Fixed Weight         | 62.8  | 64.3   | 0.977  |
> | Staged Weight        | 63.0  | 63.8   | 0.987  |
> | Dynamic Weight       | **63.2**  | **62.5**   | **1.011**  |
>
> We observe that: (1) The staged weighting scheme for the sketch-thinking reward achieves better results than the fixed-weight setting. We analyze that assigning a higher weight to the sketch-style thinking reward in the early stage of RL encourages the model to more effectively transfer the sketch-style thinking ability learned during cold-start training to the new domain of RL training data. In addition, increasing the accuracy weight toward the end of RL training helps the model refine how it utilizes sketch-style thinking to obtain correct answers. (2) The dynamic weighting scheme for the sketch-thinking reward yields even better results than the staged scheme. We attribute this to the smoother transition it provides, from emphasizing the generalization of sketch-style thinking (from cold-start to RL data) to prioritizing answer accuracy, ultimately leading to a more effective sketch-thinking ability.
>
> We have added the those discussions in Appendix E of our latest upload of revised paper (pages 15–16, lines 835–851).

---

> ### Author Response · Authors · 2025-11-21
> **Response to Reviewer FKy6 (Part 5/5)**
>
> **Comment (5)**: *"Have you run a human study on the interpretablity of the reasoning traces of the models you trained? Are the reasoning traces concise and reasonable, or they are just shorter but with many unreadable stuff?"*
>
> **Response (5)**: Thanks for your thoughtful advice. Following your comments, we conduct a human study to evaluate the interpretability of the SketchThinker-R1 reasoning traces. Specifically, we randomly find 5 human evaluators to perform the assessment. We sample 5 questions from each of MMMU, MathVision, VisuLogic, and PhyX that are correctly answered by both SketchThinker-R1 and Vanilla-R1, resulting in 20 samples in total. We present the original question, image, and the reasoning traces from both models to the evaluators. The evaluators are asked to assign a score of 0, 1, 2, 3, 4, or 5 to the reasoning traces of SketchThinker-R1 and Vanilla-R1, where a higher score indicates better interpretability. The evaluation results are as follows:
>
> | **Method**              | **Avg. Interpretability Score** |
> |---------------------|:-----------------------------:|
> | Vanilla-R1-7B       | 3.95                        |
> | SketchThinker-R1-7B | **4.25**                        |
>
> SketchThinker-R1-7B achieves a higher interpretability score than Vanilla-R1. Sketch-style reasoning is concise, and clearly presents the key logical steps for problem solving, making it much easier to follow than long, verbose reasoning traces. As a result, SketchThinker-R1 attains better interpretability scores than Vanilla-R1.
>
> Since the human study is based on a small sample size, we further conduct a large-scale LVLM-based evaluation of the interpretability of the reasoning traces produced by our model. Specifically, we utilize Qwen3-VL-Plus as the evaluator. We collect all samples from the four evaluation benchmarks (MMMU, MathVision, VisuLogic, PhyX) where both SketchThinker-R1-7B and Vanilla-R1-7B generate the correct answer. We then prompt Qwen3-VL-Plus to assign a score in range [0, 5.0] to the reasoning traces of both SketchThinker-R1-7B and Vanilla-R1-7B, where a higher score indicates better interpretability. The results are as follows:
>
> | **Method**              | **Avg. Interpretability Score** |
> |---------------------|:-----------------------------:|
> | Vanilla-R1-7B       | 4.12                        |
> | SketchThinker-R1-7B | **4.33**                        |
>
> The LVLM evaluation results are consistent with the human study, showing that our method achieves higher interpretability scores than Vanilla-R1. This further demonstrates the high interpretability of the reasoning traces produced by our trained model.
>
> We also provide qualitative examples of the reasoning traces from SketchThinker-R1 and Vanilla-R1. Please refer to Fig.3, Fig.5, Fig.6, Fig.7, and Fig.8.
>
> We have added the those discussions in Appendix G of our latest upload of revised paper (pages 18, lines 932–959).
>
> ---
>
> We hope that our responses have addressed your concerns. If you have additional comments or concerns, please let us know and we will be more than happy to answer.
>
> Best,
>
> Authors

---

### Author Response · Authors · 2025-11-21
**General Response**

Dear AC and Reviewers,

We are sincerely grateful to all of you for dedicating your time and effort to providing these detailed and thoughtful reviews, which have helped us improve the quality of our paper. We have carefully revised the paper based on your feedback. For your convenience, we have highlighted all revisions made compared to the initial version in blue.

**This work receives all positive recommendations with scores of [6, 6, 6] and confidence of [4, 4, 4].** Here, in addition to the point-by-point responses to each reviewer, we would like to summarize the contributions of this work and highlight the new results added during the rebuttal phase.

---

We are delighted that the reviewers appreciate and recognize the following strengths and contributions of this work:

- The motivation of reducing token number is reasonable for multimodal tasks thinking. This work demonstrates a robust solution to the critical challenge of high computational inference costs in advanced multimodal models. **[ag5r, 5YsX]**

- The three‐stage pipeline (cold‐start conversion of reasoning style, reward model, RL) is reasonably well‐designed, novel, comprehensive, and makes sense, which moves beyond superficial reasoning length constraints by directly fostering an "intrinsic sketch-style thinking" within large multimodal models. **[All Reviewers]**

- Experiments consistently show good results spanning different domains. The method gives good efficiency gains on multimodal reasoning, with 64% token reduction, while keeping the accuracy. **[All Reviewers]**

- This paper benefits from clear writing, well-structured sections. The three‐stage pipeline is coherently described. **[ag5r, 5YsX]**

Below, we summarize the main concerns from reviewers and our responses.

| **Dimension** | **Reviewer** | **Key Concerns** | **Our Main Actions** |
|-|-|-|-|
| Scaling Ability | FKy6 | Is the proposed method scalable and stable with larger training set and more training steps? | Conduct **scaling experiments** with respect to both the size of the RL training set and the number of RL training steps, and observe that **SketchThinker-R1 scales well along those two axes and consistently outperforms Vanilla-R1**. |
| Reward Model Design | FKy6, ag5r | More analysis and ablation on SketchJudge reward model design could make the paper stronger. | Provide **additional analysis on the design of the SketchJudge reward model**, including ablation on weight of sketch-style thinking reward, ablation between dense reward and binary reward, and strategies for combining sketch-style thinking reward with accuracy and format rewards. |
| Cold-Start Data Quality | 5YsX | Are the generated sketch-style reasoning data for cold-start encapsule all necessary steps from original reasoning process? | Assess the quality of our generated sketch-style reasoning data via **human study, LVLM-based evaluation, and case study**. Our generated sketch-style reasoning **receives high quality score from both human and LVLM evaluation**. Case study also illustrates the generated sketch-style reasoning effectively captures all key steps in original reasoning. |
| Details of Baselines | ag5r, 5YsX | Good to include details of baseline training data. Comparison with a baseline optimized for absolute maximal accuracy. | Provide **additional details** on the training data of our adopted efficient reasoning baselines. Establish **an R1-style baseline optimized for absolute maximum accuracy** and compare its performance with SketchThinker-R1, and observe that SketchThinker-R1 still outperforms this baseline. |
| Interpretability of Sketch-Style Reasoning | FKy6 | Are the sketch-style reasoning traces from SketchThinker-R1 easy to understand and reasonable? | Evaluate the interpretability of the reasoning traces of SketchThinker-R1 through both **human study and large-scale LVLM-based evaluation**, and observe that sketch-style reasoning receives higher interpretability score compared to long, verbose reasoning from Vanilla-R1. |

---

Again, thank you for your time in reviewing our work! If you have any additional comments or concerns, please let us know and we will be more than happy to answer.

Best,

Authors

---

### Meta-Review · Area_Chair_uTaw · 2025-12-27

**Summary:**

This paper proposes SketchThinker-R1, a method inspired by human sketch-style reasoning to enhance inference efficiency in large multimodal models (LMMs). The framework comprises three core stages: Sketch-Mode Cold Start (converting long reasoning into concise sketch-style reasoning and finetuning the base model), training a SketchJudge Reward Model (to evaluate and prioritize sketch-style reasoning), and Sketch-Thinking Reinforcement Learning (to generalize the ability). Experimental results show over 64% reduction in reasoning token cost across four benchmarks without sacrificing answer accuracy, with qualitative analysis confirming focus on key problem-solving cues.

Reviewers’ core concerns informing the decision included:
* The transparency and interpretability of the Sketch-Style reasoning traces;
* The details of experiments, such as how the hyperparameters are set for both the proposed method and baselines, and more ablation studies.
* The scalability of the proposed method and how much time (GPU hours) in practice could be saved.
* The assessment of the cold-start data quality.

Overall, reviewers consistently have relatively positive scores, and most concerns are addressed following the rebuttal. Since the merits of this borderline paper outweigh the flaws, I recommend acceptance.

**Reviewer Concerns:**

Most of the common concerns are addressed by the authors' rebuttal.

**Reviewer Scores:**

Though none of the reviewers replied during the discussion, this submission received borderline ratings before the discussion, and the authors have addressed most of their concerns point-by-point during the discussion.

---

### Decision · Program_Chairs · 2026-01-26

Accept (Poster)